# Upstream Stimulatory Factors Regulate HIV-1 Latency and Are Required for Robust T Cell Activation

**DOI:** 10.3390/v15071470

**Published:** 2023-06-28

**Authors:** Riley M. Horvath, Ivan Sadowski

**Affiliations:** Molecular Epigenetics Group, Department of Biochemistry and Molecular Biology, LSI, University of British Columbia, Vancouver, BC V6T 1Z3, Canada; riley.mhorvath@gmail.com

**Keywords:** USF1, USF2, HIV-1, latency, LTR, RBF-2, transcription, T cell activation, immune, inflammation, epigenetics

## Abstract

HIV-1 provirus expression is controlled by signaling pathways that are responsive to T cell receptor engagement, including those involving Ras and downstream protein kinases. The induction of transcription from the HIV-1 LTR in response to Ras signaling requires binding of the Ras-responsive element binding factor (RBF-2) to conserved *cis* elements flanking the enhancer region, designated RBE3 and RBE1. RBF-2 is composed minimally of the USF1, USF2, and TFII-I transcription factors. We recently determined that TFII-I regulates transcriptional elongation from the LTR through recruitment of the co-activator TRIM24. However, the function of USF1 and USF2 for this effect are uncharacterized. Here, we find that genetic deletion of *USF2* but not *USF1* in T cells inhibits HIV-1 expression. The loss of USF2 caused a reduction in expression of the USF1 protein, an effect that was not associated with decreased *USF1* mRNA abundance. USF1 and USF2 were previously shown to exist predominately as heterodimers and to cooperatively regulate target genes. To examine cooperativity between these factors, we performed RNA-seq analysis of T cell lines bearing knockouts of the genes encoding these factors. In untreated cells, we found limited evidence of coordinated global gene regulation between USF1 and USF2. In contrast, we observed a high degree of genome-wide cooperative regulation of RNA expression between these factors in cells stimulated with the combination of PMA and ionomycin. In particular, we found that the deletion of *USF1* or *USF2* restricted T cell activation response. These observations indicate that USF2, but not USF1, is crucial for HIV-1 expression, while the combined function of these factors is required for a robust T cell inflammatory response.

## 1. Introduction

Despite decades of research, the etiological agent of acquired immunodeficiency syndrome (AIDS), the human immunodeficiency virus (HIV-1), remains a pressing global healthcare issue as nearly 40 million individuals are infected globally and close to a million deaths occur annually [1]. Current antiretroviral therapy (ART) typically limits HIV-1 replication in most individuals to a chronic asymptomatic infection [2]. However, ART treatment must be maintained for the lifespan of infected individuals to prevent a rebound of viral replication from the latently infected cells [3], and often altered due to the emergence of drug resistant HIV-1 variants [4]. The barrier to a cure for HIV-1/AIDS is the population of infected CD4^+^ T memory cells that establish proviral latency during early infection. This extremely long-lived latently-infected cell population provides a reservoir of viremia that evades host immune surveillance and is invulnerable to ART [4,5]. Various proposed strategies directed towards eliminating latently-infected cells involve modulating the expression of latent provirus [3]. Often referred to as “shock and kill” and “block and lock”, these strategies employ latency reversing and latency promoting agents, respectively, and have been the subject of intense recent investigation [6,7]. However, initial trials using latency reversing agents in patients during antiretroviral therapy did not significantly reduce the pool of latently infected CD4^+^ T cells in vivo [3,8]. Consequently, it is recognized that treatment with latency modulating agents must be applied with consideration of effects on the global immune response, including the function of CD8^+^ T cells [9,10]. Overall, the development of effective therapies that are based on modulating provirus expression requires a more detailed understating of the mechanisms underpinning HIV-1 latency.

The 5′ long terminal repeat (LTR) serves as the HIV-1 promoter and possesses numerous *cis* elements that bind host cell transcription factors [11]. Multiple *cis* elements within the 5′ LTR bind transcription factors that are regulated downstream of T cell receptor signaling, causing provirus expression to be tightly linked to CD4 receptor engagement [12]. The Ras-responsive factor binding elements three and one (RBE3 and RBE1) are highly conserved on the HIV-1 provirus in patients who develop AIDS [13] (Figure 1A) and were initially characterized as required for induction of the HIV-1 LTR in response to Ras and MAPK signaling [14]. These conserved elements bind the RBF-2 complex that is comprised minimally of the TFII-I, USF1, and USF2 proteins [15]. Additionally, RBF-2 was found to be associated with Yin Yang 1 (YY1) at the upstream RBE3 element, which plays a role in establishing and enforcing latency [16]. Mutations of RBE1 and RBE3 abrogate the binding of RBF-2 in vivo and render the provirus incapable of transcriptional induction in response to T cell signaling [15,17,18,19]. This effect is mediated at least partially by the recruitment of the co-factor TRIM24 to the HIV-1 LTR by TFII-I, resulting in enhanced transcriptional elongation [20,21].

The upstream stimulatory factor (USF) was initially identified as a transcriptional regulator which binds an E-box of the adenovirus major late promoter (AdMLP) [22]. This factor predominately exists as a heterodimer of two structurally related helix-loop-helix leucine zipper (b-HLH-LZ) proteins of 43 kDa and 44 kDa, designated USF1 and USF2 [23]. The USF factors are ubiquitously expressed, although relative ratios in abundance vary depending on the tissue type [24,25]. USF1 and USF2 share 44% sequence identity overall, with the C-terminal b-HLH-LZ region displaying ~70% conservation [25]. Given their similarities in sequence and their effect for activation of the AdMLP [23], in addition to USF1/USF2 heterodimers predominating within cells [26], the functional differences between these proteins have largely been overlooked. However, *USF1* and *USF2* knockout mice displayed unique phenotypes, likely representing differential gene expression [27], indicating that these factors have distinct functions. USF1 and USF2 are associated with both positive and negative effects on transcription that are dependent upon the target promoter context and cellular differentiation state [27,28], where the temporal and spatial nature of the USF function generates differential effects in specific cell types [29,30,31]. The activity of the USFs has been shown to be controlled by signaling pathways through phosphorylation [32] which regulates both the DNA binding function and protein–protein interactions [33,34]. The phosphorylation of USF2 by CDK5 is thought to contribute to carcinogenesis [35], while the CK2-dependent phosphorylation of USF1 causes altered expression of genes involved in metabolism [36]. Moreover, T cell activation induced by the phorbol ester PMA causes extensive phosphorylation of both USF1 and USF2, although the regulatory implications of these modifications have not been determined [15].

USF was among the first cellular factors shown to bind the HIV-1 promoter at an E-box element located at −160 on the LAI subtype B LTR [37,38]. This upstream E-box was found to be conserved on only ~15% of LTRs from provirus in samples from HIV-1 patients who developed symptoms [13]. The effect of USF for HIV-1 expression was subsequently found to be predominately mediated as a component of RBF-2. The binding of USF1/2 to the RBE3 and RBE1 elements, positioned at −130 and −22 (Figure 1A), is stimulated by interaction with TFII-I [18,19]. Mutations within the core RBE3 element (ACTGCTGA, Figure 1A) or immediate 3’ sequences prevent reactivation of HIV-1 in response to T cell signaling, but also abrogates binding of at least four transcription factors to this region of the LTR, including USF1, USF2, TFII-I, and YY1 [15,16,18]. Consequently, the overall effect of USF1 and USF2 for the regulation of HIV-1 expression has not been clarified. Furthermore, although USF1/2 have been reported to be regulated by multiple different protein kinases [32], the role of these factors for the regulation of signal-responsive genes in T cells has not been characterized. As core components of the RBF-2 complex, we propose that the USF proteins play a critical role for regulation of HIV-1 expression, at least in part due to recruitment of TFII-I and TRIM24 to the highly conserved RBE1 and RBE3 LTR elements. Consequently, in this study, we examined the effect(s) caused by the genetic depletion of *USF1* and *USF2* on HIV-1 expression in T cells, and the effect on global gene expression in response to T cell signaling.

In this study, we demonstrate that HIV-1 provirus expression was inhibited upon the depletion of USF2, whereas the loss of USF1 had no significant effect. Additionally, the loss of USF2 resulted in a reduction in USF1 protein expression, an effect that did not correlate with *USF1* mRNA abundance. Furthermore, we found that USF1 and USF2 were key factors required for T cell activation, as the loss of either protein caused global constraint of T cell signal-induced genes. The *USF2* gene knockout limited T cell activation to a greater extent than the *USF1* knockout, and additionally rendered T cells more restrictive to HIV-1 infection. These observations clarified the role of USF1 and USF2 for regulation of HIV-1 transcription and identified these proteins as key signal regulated factors that play a vital role in activation of the T cell inflammatory response.

## 2. Materials and Methods

### 2.1. Cell and Virus Culture

Jurkat E6-1, Jurkat Tat mHIV-Luciferase, TZM-bl, and HEK293T cells were cultured under standard conditions of 37 °C and a 5% CO_2_ atmosphere. The Jurkat cells were cultured in RPMI-1640 media while HEK293T and TZM-bl were cultured in DMEM, supplemented with 10% FBS, penicillin [100 units/]mL, streptomycin [100 g/]mL, and L-glutamine [2 mM]. The vesicular stomatitis virus G (VSV-G) pseudotyped viral stocks were produced by co-transfecting HEK293T cells with a combination of viral molecular clone, psPAX, and pVSV-G at a ratio of 8 μg:4 μg:2 μg. The transfections were performed using polyethylenimine (PEI) at a ratio of 6:1 (PEI:DNA) in Gibco Opti-MEM^tm^. The lentiviral infections were performed by plating 1 × 10^6^ cells in 24-well plates with media containing 8 μg/mL polybrene and the amount of viral supernatant to give the desired multiplicity of infection (M.O.I.) as indicated. The plates were subsequently spinoculated for 1.5 h at 1500 rpm.

### 2.2. Immunoblotting

Western blotting was performed as previously described [21]. The antibodies were as follows: Tubulin (1:20,000)—Abcam ab7291, Flag (1:20,000)—Sigma Aldrich F3165, Myc (1:2500)—Santa Cruz Biotechnology sc-40, USF1 (1:500)—Santa Cruz Biotechnology sc-390027, USF2 (1:8000)—Abcam ab264330, Goat Anti-Rabbit-HRP—Abcam ab6721 (1:2,000,000), and Goat Anti-Mouse-HRP—Pierce 1858413 (1:20,000).

### 2.3. Luciferase Expression Assays

For the Jurkat Tat mHIV-Luciferase expression assays, 1 × 10^5^ of cells were plated with 100 µL media in 96-well plates. The luciferase activity was measured after the indicated time of treatment. The measurements were performed using the Superlight™ luciferase reporter Gene Assay Kit (BioAssay Systems, Hayward, CA, USA) as per the manufacturer’s instructions. The 96 well plates were read using a Victor™ X3 Multilabel Plate Reader.

### 2.4. Transfection Assays

A total of 6.67 × 10^5^ WT or *USF2* KO HEK293T cells were plated in 2 mL DMEM in six-well plates. The following day, the cells were transfected with 0.5 ng of the indicated LAI LTR-GFP reporter construct and 1 μg of the indicated EF1α-RFP construct. The transfections were performed using polyethylenimine (PEI) at a ratio of 1:4 (DNA:PEI) in Gibco Opti-MEM™. At one day post-transfection, the cells were detached using trypsin-EDTA and analyzed using flow cytometry.

### 2.5. Flow Cytometry

The cells were treated as indicated in the Figure legends. Following the indicated treatment, Jurkat-derived cells were suspended in PBS while the HEK293T-derived cells were suspended in PBS containing 10% trypsin-EDTA to prevent aggregation. Flow cytometric analysis was performed using a BD Biosciences LSRII-561 instrument where the threshold forward scatter (FSC) and side scatter (SSC) parameters were set so that a homogenous population of live cells was counted (Appendix A). FlowJo software (v10.8.1) (TreeStar, Hendersonville, TN, USA) was used to analyze the data and determine the indicated mean fluorescent intensity (MFI).

### 2.6. Generation of USF1 and USF2 Knockout Clonal Lines

The *USF1* KO and *USF2* KO Jurkat Tat mHIV-Luciferase and wildtype Jurkat E6-1 clonal cell lines were generated by CRISPR-Cas9 gene editing. A total of 2 × 10^6^ cells were co-transfected with vectors expressing Cas9 and gRNA, targeting either genomic *USF1* or *USF2*. The transfection was performed using a Neon Transfection System (Invitrogen) as per the manufacturer’s instructions with the following settings: voltage, 1350 V; width, 20 ms; pulse number, 3×. For *USF1* KO, the Jurkat cells were co-transfected with Cas9-BFP (pU6-CBh-Cas9-T2A-BFP: Addgene 64323) and gRNA (pSPgRNA: Addgene 47108) containing gRNA that targeted GCCAGGTAAGGGAGGGGGCC and GGAAGACGTACTTGACGTTG. For *USF2* KO, the Jurkat cells were co-transfected with Cas9-mCherry (pU6-CBh-Cas9-T2A-mCherry: Addgene 64324) and gRNA-BFP (pKLV2.2-h7SKgRNA-hU6gRNA-PGKpuroBFP: Addgene 72666) containing gRNA that targeted GTCGTGGCTGCCAGGGGCAC, GGAGGAGGGCGTCGAGCTGC, and GGCGGCCGAGGCTGTCAGCG. For *USF1* KO, the transfected cells were isolated by live sorting (Astrios Flow Cytometer) the BFP+ cells into 96-well plates containing RPMI-1640, while for *USF2* KO, the mCherry+/BFP+ cells were isolated. The clones were expanded, and the knockout of *USF1* and *USF2* was validated by PCR genotyping and western blotting.

To generate the *USF2* KO TZM-bl and HEK293T lines, the cells were co-transfected using polyethylenimine (PEI) with Cas9 (pU6-CBh-Cas9-T2A-BFP: Addgene 64323) and gRNA-Puro (pKLV2.2-h7SKgRNA-hU6gRNA-PGKpuroBFP: Addgene 72666) containing gRNA that targeted GGAGGAGGGCGTCGAGCTGC and GGCGGCCGAGGCTGTCAGCG. Briefly, 6.67 × 10^5^ cells were plated with 2 mL DMEM in a six-well plate. The following day, the transfection was performed using 3 μg Cas9, 1 μg gRNA-Puro, and 12 μg PEI. The transfected cells were selected using 1 μg/mL puromycin for four days, at which point single cells were plated in 96-well plates by limiting dilution. The clones were expanded and *USF2* KO was validated by western blotting.

### 2.7. USF1 and USF2 shRNA Knockdown

The Jurkat mHIV-Luciferase cells were infected with the pLKO empty vector or pLKO shRNA expressing lentivirus at a M.O.I. ~10. The shRNA transduced cells were cultured for up to 8 days with 3 µg/mL puromycin, during which pools of puromycin-selected cells were prepared for the indicated analysis. The MISSION shRNA clones (Sigma) used for the knockdown were as follows: USF1, TRCN0000020679—GCTGGATACTGGACACACTAA (3′ UTR); USF2-1, TRCN0000020734—TCCTCCACTTGGAAACGGTAT (3′ UTR); USF2-2, TRCN0000020737—TCCAGACTGTAACGCAGACAA (CDS); and TFII-I, TRCN0000019315—CGGATGAGTGTAGATGCTGTA (CDS).

### 2.8. ChIP-qPCR

ChIP-qPCR was performed as in [20,21]. The antibodies used were as follows: Myc (10 μg)—Santa Cruz Biotechnology sc-40, USF2 (10 μg)—Abcam ab264330, TFII-I (10 μg)—BD Biosciences 610942, and TRIM24 (10 μg) –Proteintech 14208-1-AP. The oligos used for the qPCR were: E-Box, Fwd 5′ GTGAGCCTGCATGGAATGGA, Rev 5′ CGGATGCAGCTCTCGGG; RBE3, Fwd 5′ AGCCGCCTAGCATTTCATC, Rev 5′ CAGCGGAAAGTCCCTTGTAG; RBE1, Fwd 5′ AGTGGCGAGCCCTCAGAT, Rev 5′ AGAGCTCCCAGGCTCAGATC; and Gag, Fwd 5′ AGCAGCCATGCAAATGTTA, Rev 5′ AGAGAACCAAGGGGAAGTGA.

### 2.9. RT-PCR

RNA was extracted from Jurkat Tat mHIV-Luciferase cells following the indicated treatment using the RNeasy Kit (Qiagen). The RNA was analyzed using the Quant Studio 3 Real-Time PCR system (Applied Biosystems) and the Power SYBR^®^ Green RNA-to-CT™ 1-Step Kit (Thermo Fisher) as per the manufacturer’s instructions. The RT-PCR data was normalized to GAPDH expression using the ΔΔCt method. The primers used were USF1 Fwd 5′ GCTCTATGGAGAGCACCAAGTC, Rev 5′ AGACAAGCGGTGGTTACTCTGC; and GAPDH Fwd 5′ TGCACCACCAACTGCTTAGC, Rev 5′ GGCATGGACTGTGGTCATGAG.

### 2.10. RNA-Seq

RNA was extracted from the wildtype, *USF1* KO, or *USF2* KO Jurkat Tat mHIV-Luciferase human T cells that were left untreated or stimulated with PMA/ionomycin for 4 h. Sample quality control was performed using an Agilent 2100 Bioanalyzer. The qualifying samples were prepped using the NEBnext Ultra ii Stranded mRNA (New England Biolabs) standard protocol. Sequencing was performed using an Illumina NextSeq 500 with Paired End 42 bp × 42 bp reads. The demultiplexed read sequences were uploaded to the Galaxy web platform [39] and aligned to the hg38 reference genome using STAR. Subsequently, the transcript assembly was accomplished using featureCounts and differential gene expression (DEG) was determined by DESeq2 analysis. The DEG was defined as having a fold change in the expression > 1.5 and *p*-value < 0.05 as to the compared condition. Volcano plots were generated on the Galaxy web platform and gene expression heatmaps were created in RStudio.

### 2.11. Statistics and Reproducibility

All the replicates represent independent biological samples, and the results are presented as mean values  ± the standard deviation, as shown by error bars. The number of times an experiment was performed is indicated in the Figure legends. The *p*-values were determined by performing unpaired sample *t*-tests with the use of GraphPad Prism 9.0.0. The statistical significance is indicated as * *p* < 0.05, ** *p* < 0.01, or *** *p* < 0.001, with n.s. denoting a non-significant *p* ≥ 0.05.

## 3. Results

### 3.1. Regulation of the HIV-1 Expression by USF1 and USF2

The 5′ LTR serves as the HIV-1 promoter and contains several *cis* elements that bind USF1/2. The highly conserved RBE1 and RBE3 elements are both occupied by the RBF-2 complex that is minimally composed of USF1/USF2 and TFII-I [19]. Additionally, USF heterodimers bind an upstream E-Box present on the prototypical LAI LTR (Figure 1A) [15]. Previous studies implicated USF1 and/or USF2 as regulators of HIV-1 expression, mostly based on transient transfection assays and LTR mutations that abolish binding of these factors [15,19,38,40,41]. However, interpretation of effects caused by mutations of the RBE1 and RBE3 elements is complicated by the fact that they also impair binding of at least TFII-I and YY1 [16,18,19]. Consequently, to examine the specific effects of USF1 and USF2 on HIV-1 expression, we produced *USF1* and *USF2* gene knockouts (KO) (Figure 1C,D) in the Jurkat Tat mHIV-Luciferase T cell line bearing an integrated HIV-1 mini-virus where luciferase expression is driven by the 5′ LTR (Figure 1B) [42]. Surprisingly, we noted that USF1 protein abundance was reduced in all the *USF2* knockout lines (Figure 1D), whereas USF2 protein expression was not affected by the *USF1* knockout (Figure 1C). We examined the activation of HIV-1 expression in these knockout lines and found that the *USF1* disruption tended to cause a slight increase in luciferase expression in unstimulated as well as activated conditions, as mediated by the T cell signaling agonist PMA (Figure 1E). In contrast, provirus expression was diminished in both unstimulated and stimulated conditions in all the *USF2* knockout lines (Figure 1F). Consistent with results using the knockout lines, we found that shRNA mediated depletion of USF1 (Appendix A) had no effect on HIV-1 provirus expression (Appendix A), while shRNA knockdown of USF2 (Appendix A) inhibited HIV-1 expression in both untreated cells and cells stimulated with the phorbol ester PMA (Appendix A). Additionally, we found that TZM-bl cells were refractory to HIV-1 induction upon USF2 depletion (Appendix A), confirming the requirement of USF2 for the full expression of chromosomally integrated HIV-1.

As USF1 protein expression was dependent upon USF2 in Jurkat T cells, we examined the effect of the *USF2* gene knockout in HEK293T cells where we also observed a correlation between USF1 and USF2 protein expression (Figure 2A). Next, we used an LAI-derived LTR-Tat-IRES-GFP reporter construct to assess the effect of the *USF2* knockout on HIV-1 expression in these cells (Appendix A). Consistent with the above results, we observed significantly lower GFP expression in the HEK293T *USF2* KO cells relative to the wildtype (WT) (Figure 2B). Remarkably, we found that co-transfection of the HEK293T *USF2* KO cells with vectors expressing either USF1 or USF2 (Figure 2C) and the HIV-1 LTR-Tat-IRES-GFP reporter not only rescued LTR transcription but also caused a greater activation of GFP expression than that in wildtype cells (Figure 2D and Appendix A). These observations suggest that any combination of USF1 and USF2 can activate transcription from the HIV-1 LTR, but because USF1 protein expression was decreased in the *USF2* knockout lines while the USF2 expression was unaffected by the USF1 depletion, only the *USF2* knockouts produce a significant effect on HIV-1 LTR transcription. Of note, the knockout of either *USF1* or *USF2* was not associated with any noticeable change in the cell phenotype, viability, or growth (Appendix A).

### 3.2. Effect of USF1 and USF2 on Response of HIV-1 to Latency Reversing Agents and Tat

We examined the effect of the *USF1* and *USF2* knockouts on reactivation of HIV-1 in response to various mechanistically diverse latency reversing agents (LRAs). In T cells, the phorbol ester PMA activates protein kinase C (PKC) and Ras-MAPK signaling, while ionomycin causes the release of intracellular calcium, which activates calcineurin. Consequently, treatment with the combination of PMA and ionomycin mimics T cell activation produced by CD4 receptor engagement [43]. We found that *USF2* KO inhibited HIV-1 provirus activation in response to PMA or ionomycin, individually or in combination (Figure 3A). In contrast, *USF1* KO caused a slight increase in response to the combination of PMA/ionomycin or PMA alone (Figure 3A). The effect of the *USF1* and *USF2* knockouts in response to PMA and ionomycin was most pronounced at 6 h post-treatment (Figure 3B). As observed previously, we found that HIV-1 expression became exhausted upon prolonged treatment (20 h) with PMA/ionomycin in WT cells [15,17], and this effect was not significantly different in the *USF1* and *USF2* knockout lines (Figure 3B). Interestingly, the loss of USF1 caused slightly decreased induction upon treatment with ionomycin alone (Figure 3A), indicating that USF1 was not required for HIV-1 reactivation in response to PKC and Ras-MAPK signaling but may play a role for activation by calcium–calcineurin signaling. Ingenol 3-angelate (PEP005) also activated PKC signaling [44], and similar to the effect of PMA treatment, we observed a significantly lower response to PEP005 in the *USF2* KO cells but no effect in the *USF1* KO line (Figure 3A). These results indicate that USF2 is required for reactivation of HIV-1 in response to the PKC, Ras-MAPK, and calcineurin T cell signaling pathways, whereas USF1 may cause some inhibition in response to MAPK activation.

The latent HIV-1 provirus can be reactivated independently of T cell signaling by altering the epigenetic composition or interaction of bromodomain-containing proteins with the LTR. Treatment with the histone deacetylase inhibitor (HDACi) suberanilohydroxamic acid (SAHA) [45] modestly increased HIV-1 expression, an effect that was limited by USF1 depletion and almost completely abrogated by the *USF2* knockout (Figure 3A). We also examined the effect of the *USF* knockouts on reactivation in response to the bromodomain inhibitors JQ1 and IACS-9571. JQ1 is a well-studied LRA that inhibits the BRD4 bromodomain, causing increased Tat transactivation [46], while IACS-9571 facilitates transcriptional elongation by targeting TRIM24 to the LTR [20,21]. We found that the *USF2* knockout significantly limited the effect of JQ1, whereas *USF1* KO had no effect (Figure 3A). The *USF2* knockout, and to a lesser extent *USF1* KO, inhibited reactivation of HIV-1 expression in cells treated with IACS-9571 (Figure 3A), suggesting that both USFs impact regulation of HIV-1 by TRIM24.

We also examined the effect of the *USF2* gene knockout on activation of HIV-1 expression by the viral transactivator protein Tat. For this experiment, we transfected the *USF2* knockout HEK293T lines (Figure 2A) with HIV-1 LTR-Tat-IRES-GFP (+Tat) or LTR-GFP reporter plasmids (-Tat) (Appendix A). We observed that GFP expression from both the HIV-1 LTR reporter plasmids was significantly lower in the *USF2* knockout cells compared to the WT (Figure 3C and Appendix A). These observations indicate that full activation of HIV-1 transcription by Tat requires USF2 protein, although USF2 also has a significant effect on HIV-1 expression in the absence of Tat.

### 3.3. USF2 Stabilizes USF1 Protein

We observed that depletion of USF2 in Jurkat T cells (Figure 1D and Appendix A) or HEK293T cells (Figure 2A) invariably caused a corresponding decrease in USF1 protein abundance. To examine the mechanism by which USF1 protein abundance was dependent upon USF2, we generated a Jurkat *USF2* KO cell line where Flag-tagged USF2 was expressed using a doxycycline-inducible promoter. We found that doxycycline-induced expression of USF2-Flag in this *USF2* KO cell line rescued the expression of USF1 protein to greater levels compared to WT cells (Figure 4A). We also observed that *USF2* KO did not cause a decrease in *USF1* mRNA abundance compared to WT cells (Figure 4B), regardless of the T cell activation state (Figure 4C), indicating that differences in USF1 expression in *USF2* KO cells must relate to protein stability or effects on mRNA translation. Furthermore, stimulation with PMA/ionomycin did not alter USF1 or USF2 protein abundance in Jurkat T cells, and USF1 protein levels were decreased in the *USF2* knockout cells regardless of the cellular activation state (Figure 4D).

### 3.4. Effect of USF Knockouts on the Interaction of RBF-2 Proteins with the LTR

We examined the requirements of USF1 and USF2 for binding to the HIV-1 LTR using chromatin immunoprecipitation and quantitative PCR (ChIP-qPCR) analysis (Figure 5A). Since we were unable to immunoprecipitate USF1 with antibodies against native protein, we generated mHIV-Luciferase Jurkat *USF1* and *USF2* KO cell lines that stably express Myc-tagged USF1 (Figure 5B). We found that USF1-Myc was associated with the LTR in both the *USF1* and *USF2* KO cell lines (Figure 5C). However, we did observe slightly higher amounts of USF1-Myc associated with all three sites on the LTR in the *USF1* knockout line (Figure 5C), which expressed endogenous USF2 and has slightly more USF1-Myc expression than the *USF2* null cells (Figure 5B). Similarly, we found that USF2 was associated with the LTR at equivalent levels in both the wildtype and *USF1* knockout cells (Figure 5D). Furthermore, PMA/ionomycin mediated T cell activation had no effect on USF2 binding at RBE3 or RBE1 (Figure 5E), which is consistent with the previous observation that TFII-I is constitutively associated with the HIV-1 LTR [20]. These results indicate that USF1 and USF2 are both capable of binding specific *cis* elements on the LTR as homodimers in vivo, a result that is consistent with binding of recombinant USF1 and USF2 proteins in vitro [15,18]. Thus, although USF1 and USF2 were predominately shown to function as heterodimers in vivo [24,25], USF2 is capable of binding independently of USF1 at multiple sites on the HIV-1 LTR for the activation of transcription. In contrast, as indicated above, USF1 protein abundance is dependent upon USF2, and therefore is incapable of regulating HIV-1 transcription in the *USF2* knockout cells.

Next, we sought to examine requirements of the RBF-2 proteins for binding to the HIV-1 LTR. We used lentiviral-delivered shRNA to knockdown the expression of TFII-I (*GTF2I*) in the Jurkat mHIV-Luciferase cell line (Figure 1B), where we observed a partial reduction in TFII-I protein (Figure 6A). Consistent with previous results [20], even with a partial knockdown of TFII-I, we observed reduced expression of HIV-1 provirus in both the unstimulated and stimulated cells (Figure 6B), an effect that was similar in the *USF2* KO cells (Figure 1F). Interestingly, we observed no significant change in the binding of USF2 to the RBE elements in cells with a partial TFII-I knockdown (Figure 6C). In contrast, binding of TFII-I to the LTR was completely abrogated in the *USF2* KO cells (Figure 6D). Similarly, we also observed a complete loss of the co-activator TRIM24 at the LTR in the *USF2* KO cells (Figure 6E), which is consistent with our observation that this factor is recruited to the LTR by TFII-I [20]. Collectively, these results indicate that USF proteins are required for RBF-2 complex formation in vivo, including the recruitment of TFII-I to the LTR.

### 3.5. Global Regulation of Transcription by USF1 and USF2 in Unstimulated T Cells

Given the surprising observation that Jurkat T cells with *USF1* and *USF2* gene knockouts have distinct effects on the HIV-1 expression, we sought to determine the degree to which these factors cooperatively regulate global gene expression. To examine this, we performed RNA-seq on cell lines bearing *USF1* or *USF2* gene knockouts. In this analysis, we identified 236 differentially expressed genes (DEG) upon *USF1* depletion and 563 DEG following *USF2* disruption (Figure 7A,B). Of the genes affected by the loss of USF1, slightly more were downregulated than upregulated (146 to 90, respectively) (Figure 7A), while the USF2 depletion primarily resulted in decreased expression of genes (410 downregulated to 153 upregulated) (Figure 7B). Comparison of transcriptomes from the *USF1* and *USF2* KO cells demonstrated some overlap amongst the downregulated genes with 34% of the *USF1* KO downregulated genes similarly regulated in the *USF2* KO cells (Figure 7C). However, we observed only 7% of upregulated genes upon USF1 depletion that were similarly upregulated in the *USF2* knockout cells (Figure 7D). Overall, the *USF1* and *USF2* knockouts caused a surprisingly small overlap in differential gene regulatory effects in unstimulated Jurkat T cells (Figure 7E). Furthermore, gene set enrichment analysis (GSEA) indicated only one pathway with less than 25% false discovery rate (FDR) that was shared between the untreated *USF1/2* KO cells, “TNFA signaling via NFκB”, for which the target genes were downregulated (Figure 7F,G). These results indicated that USF1 and USF2 share regulatory effects for a minor subset of genes in unstimulated T cells.

### 3.6. USF1 and USF2 Cooperatively Regulate Global Gene Expression in Activated T Cells

USF1 and USF2 were reported to be signal-regulated transcription factors [33,34,36,47] that are phosphorylated upon T cell activation [15]. Consequently, we also examined global gene expression in the *USF1* and *USF2* knockout lines using RNA-seq from cells stimulated with PMA and ionomycin co-treatment. This analysis revealed a greater effect on global expression in the stimulated cells compared to unstimulated cells, with 1846 and 2667 DEG upon loss of USF1 and USF2, respectively (Figure 8A,B). Importantly, depletion of either USF1 or USF2 resulted in the majority of DEG having decreased expression rather than increased expression (Figure 8A,B). These results support the contention that USF1 and USF2 are T cell signal regulated factors that predominately behave as transcriptional activators.

Additionally, we observed a significantly greater proportion of genes that were similarly affected by *USF1* KO and *USF2* KO in activated T cells. In stimulated cells, we found that 68% of the *USF1* KO downregulated and 38% of *USF1* KO upregulated genes were similarly mis-regulated upon *USF2* KO (Figure 8C,D). Furthermore, a minority of the USF1 DEG were found to be mis-regulated opposite that of the *USF2* null cells (Figure 8E). We performed GSEA analysis on the *USF1* and *USF2* KO T cell transcriptomes to determine the predominant affected pathways, which revealed genes downstream of “TNFA signaling via NFκB”, “hypoxia”, and “interferon gamma response” as being repressed (Figure 8F), while genes associated with “oxidative phosphorylation” and “Myc targets V1” were upregulated in both the *USF1* KO and *USF2* KO cell lines (Figure 8G). Although genes from multiple pathways were similarly affected in the *USF1* and *USF2* KO lines, we observed an asymmetrical regulation between these factors for several pathways, including “mTORC1” and “Notch signaling”, which were downregulated solely in the *USF1* and *USF2* knockout cells, respectively (Figure 8F). Interestingly, USF1 was previously implicated in regulating the UV and tanning response [47,48,49]. However, we found that genes involved in “UV response down” were repressed in cells that were deficient in either USF1 or USF2 (Figure 8F), suggesting that both factors mediate this process. Taken together, these results indicate that USF1 and USF2 display a high degree of cooperativity for global gene regulation during T cell activation, but that specific subsets of genes appear to be regulated independently by these factors. As noted previously, the loss of USF2 results in a corresponding decrease in USF1 expression, and therefore it is possible that USF2 may partially compensate for the loss of USF1 protein in both unstimulated and stimulated T cells.

### 3.7. USF1 and USF2 Facilitated T Cell Activation

Having observed that loss of either USF1 or USF2 generated an abundance of downregulated genes upon T cell activation relative to wildtype cells, we examined the potential role of these factors for regulation of the inflammatory response. From the RNA-seq analysis, we identified 2790 downregulated and 3481 upregulated genes in response to T cell activation (Figure 9A). Notably, the genes that were induced in response to stimulation displayed a much greater range of expression compared to genes that were inhibited (Figure 9A). We then examined the effect of the *USF1* and *USF2* gene knockouts on expression of genes that were activated or repressed upon T cell stimulation. Here, we found that 755 (22%) and 1114 (32%) genes induced upon T cell stimulation showed significantly lower expression in the absence of USF1 and USF2, respectively (Figure 9B). The genes that were repressed upon T cell activation were much less affected by the *USF* disruption, with only 187 (7%) and 272 (10%) displaying greater expression upon the loss of USF1 or USF2, respectively (Figure 9B). These results are consistent with the above observations, indicating the role of USF1 and USF2 as transcriptional activators.

T cell activation was previously shown to alter the expression of numerous genes with a wide degree of variance [50]. This is consistent with our results indicating that nearly 50% of genes induced upon stimulation showed less than a three-fold change in their expression (Figure 9C). Consequently, it is possible that genes displaying a large variance in induction are regulated by different mechanism(s). To examine this with respect to USF1 and USF2, we segmented the genes induced upon T cell activation into three groups by the level of induction, designated low (fold change in expression between 1.5 and three), mid (fold change in expression between three and eight), and high (fold change greater than eight) (Figure 9C). By stratifying the genes in this way, it became apparent that genes that were highly expressed in response to T cell stimulation were more potently inhibited in the *USF1* or *USF2* KO cells (Figure 9D). Furthermore, we found that more of the genes that were downregulated upon USF1 or USF2 depletion were highly induced in response to T cell activation (Figure 9E). Additionally, we found that the *USF1* and *USF2* knockout cells displayed a partially defective activation of genes downstream of key T cell signaling pathways, including TNFα and Ras (Appendix A). Collectively, these results indicate that USF1 and USF2 are signal-regulated factors required for full induction of genes upon T cell activation, a finding that is consistent with the evolved capability of the HIV-1 LTR to appropriate transcription factors downstream of T cell signaling [51].

### 3.8. USF2 Deficient Cells Are Refractory for HIV-1 Infection

HIV-1 infection of resting CD4^+^ T cells is inefficient compared to infection of activated cells [52,53,54]. Although Jurkat cells are actively dividing, they too are preferentially infected when activated [55]. As HIV-1 favors the infection of stimulated T cells, and that the loss of USF1 and USF2 dampens T cell activation, we sought to examine the role of these factors for HIV-1 infection. For this purpose, we created *USF1* and *USF2* knockouts in Jurkat T cells that did not contain a previously integrated provirus. Consistent with the results shown above, we observed the loss of USF1 expression in both the *USF1* and *USF2* knockout lines (Figure 10A). These cell lines were infected with the Red-Green-HIV-1 (RGH) [56] or HIV_GKO_ [57] dual-reporter viruses where GFP is expressed from the 5′ LTR and the internal constitutive promoters express mCherry or mKO2 in RGH and HIV_GKO_, respectively (Figure 10B). These reporter HIV-1 derivatives allow for the differentiation of productive and latent infections using flow cytometry (Figure 10C,D) [56]. We found that infection of the *USF1* knockout cells produced a small but not significant decrease in RGH (Figure 10E) or HIV_GKO_ (Figure 10F) infectivity compared to the wildtype, as determined 4 days post-infection. In contrast, the *USF2* knockout cells were infected at a rate roughly half that of the wildtype cells by either reporter virus (Figure 10C–F). Additionally, infection of the *USF* knockout Jurkat Tat mHIV-Luciferase cell lines (Figure 1C,D) with a Red-Blue-HIV-1 (RBH) reporter virus similarly indicated that USF2 depletion caused the cells to be more resistant to infection (Appendix A). Interestingly, although USF2 was required for reactivation of latent provirus, we found that *USF2* KO did not affect the proportion of cells that established latency four days following infection (Figure 10G,H). Taken together, these observations indicate that USF2 is important for infection of T cells by HIV-1 through a process that may involve the cellular activation state.

### 3.9. USF2 Promoted Productive HIV-1 Infection

We found that the *USF2* knockout inhibited reactivation of latent HIV-1 provirus (Figure 1) but did not affect the proportion of latently infected cells at four days post-infection (Figure 10G,H). To further examine the role of USF for regulating latency, we monitored the proportion of productively infected cells for three weeks following infection of wildtype, *USF1* KO, and *USF2* KO cells. Previous observations indicated that the internal constitutively expressed mCherry and mKO2 reporters in cells infected with RGH and HIV_GKO_ becomes silenced following extended culture [56,57]. Consequently, we isolated the infected populations using fluorescent activated cell sorting (FACS) two days post-infection (Figure 11A). Then, the WT, *USF1* KO, and *USF2* KO infected cells were analyzed using flow cytometry over the following 22 days (Figure 11B,C). Similar to the results with unsorted infected cells (Figure 10), we found that the WT, USF1, and USF2 deficient cells produced a nearly equivalent proportion of productively infected cells at 3 and 4 days post-infection (Figure 11D,E). However, after 4 days of infection, the *USF2* knockout cells accumulated latent HIV-1 infections at a higher proportion compared to the WT or *USF1* KO cells (Figure 11D,E). These results indicate that USF2 promotes the expression of HIV-1 and encourages productive replication upon infection.

## 4. Discussion

The upstream stimulatory factors (USFs) were among the first proteins identified as sequence-specific DNA binding factors required for activation of transcription in eukaryotes [22,24], and among the first human factors discovered to bind the 5′ HIV-1 LTR [38]. Previous observations indicated that USF1 and USF2 regulate transcription in response to multiple signaling pathways involving protein kinases, including at least MAPK/ERKs [32,58], casein kinase 2 (CK2) [34], CDK2 [59], protein kinase C (PKC) [60], DNA-dependent protein kinase (DNAPK) [61,62], protein kinase A (PKA) [63], and glycogen synthase kinase-3 (GSK-3) [62]. These protein kinases represent a significant fraction of the signaling networks in human cells, which implicate USF1 and USF2 as key regulatory factors involved in coordinating gene expression in response to diverse cellular stimuli. USF1 and USF2 are ubiquitously expressed in human cells [25], and consequently, regulation of their function is likely cell-type specific. However, mechanistic details regarding regulation of USF1/2 function by signaling mechanisms are sparse. Nevertheless, our results are consistent with these observations, and indicate that these factors play a significant role in the regulation of the signal-responsive gene expression during T cell activation. USF2, in particular, is required for the induction of a significant proportion of highly induced genes in response to T cell signaling (Figure 9).

USF1/2 were found to bind the prototypical LAI HIV-1 LTR at an E-box element in the upstream modulatory region at −160 to −165 from the transcriptional start site [63], where it was shown to activate transcription in cooperation with Ets-1 [41]. This *cis* element is not well conserved on LTRs from infected individuals who developed symptoms of AIDS [13]. USF1 and USF2 were subsequently found to form part of the factor RBF-2, which binds two highly conserved *cis* elements at −130 (RBE3) and −20 (RBE1) [15,19]. The RBE3/1 sites were initially identified due to their requirement for activation of HIV-1 transcription by activated Ras in T cells [14]. The RBF-2 complex was subsequently purified from Jurkat nuclear extracts using oligonucleotide affinity [64], and was found to contain USF1, USF2, and TFII-I [15]. The RBE3 element is not typical of other characterized binding sites for USF1/2 (ACTGCTGA, Figure 1A), and the interaction of USF with this site in vitro is stimulated upon addition of TFII-I [15,18]. In contrast, the RBE1 site (CAGCTGC) [14] positioned near the transcriptional start resembles a typical E-box element (CANNTG, Figure 1A), and consequently, binding of USF1/2 to this site is less dependent upon TFII-I [19]. Mutations of these elements significantly inhibit activation of HIV-1 expression in response to T cell receptor cross linking or the combination of PMA and ionomycin [15,18,19]. Consistent with these previous observations, we found that the disruption of *USF2* inhibited the expression of HIV-1 in response to T cell signaling as well as a variety of latency reversing agents (Figure 1 and Figure 3). Furthermore, the *USF2* disruption limited the initial incidence of HIV-1 infection (Figure 10) and reduced the proportion of infected cells that underwent productive gene expression (Figure 11). We note that our analysis of the effects of *USF1* and *USF2* knockouts were predominately examined in the Jurkat T cell line, and it will be important to examine effects caused by depletion of these factors in normal CD4^+^ T cells.

The mechanism by which the factors bound at RBE3 and/or RBE1 cause activation in response to T cell signaling has not been elucidated. However, TFII-I, in association with the serum response factor (SRF), was shown to regulate expression of the *c-fos* promoter in response to Ras-stimulated MAPK signaling by direct interaction with ERK1 [65,66]. Stimulation of T cells by antigen presentation causes activation of multiple pathways downstream of the T cell receptor, including the Ras-Raf-Mek-Erk (MAPK) pathway [17]. Consequently, the role of USF1/2 for activation of HIV-1 transcription likely includes, at least in part, interaction with TFII-I for binding the conserved RBE1 and RBE3 sites on the LTR [18]. Consistent with this possibility, we observed loss of TFII-I binding to the LTR in cells with the *USF2* disruption (Figure 6). Additionally, as mentioned above, USF1 and USF2 were reported to be regulated by multiple protein kinases, and it seems likely that these proteins mediate additional function(s) to regulate HIV-1 activation in response to T cell stimulation. Accordingly, we found that USF1, USF2, and TFII-I become hyperphosphorylated in T cells stimulated with PMA and ionomycin [15]. Interestingly, however, we note that in *USF2* knockout cells, the HIV-1 provirus remained significantly responsive to the PKC agonists PMA and PEP005 (Figure 1F and Figure 3A). This indicates that additional factors capable of binding to the LTR, potentially including AP1, GABP, and Ets [11], may at least partially compensate for the loss of RBF-2 components, demonstrating a particular resilience of this viral promoter for maintaining coordinate regulation with T cell signaling. Despite that USF1 and USF2 are thought to predominately exist as heterodimers in most cell types [25,27], we found that disruption of their genes caused significantly different alterations in the transcriptome and where we observed a greater regulatory overlap in stimulated cells relative to unstimulated cells (Figure 7 and Figure 8). Interestingly, we note distinct factor-specific effects on genes involved in mTORC1 and Notch signaling in activated T cells (Figure 8F). This is consistent with previous observations indicating discrete functions for USF1 and USF2. For example, USF2 was associated with progression of leukemia through upregulation of *HOXA9*, but disruption of *USF1* alone had minimal effects [31]. Overexpression of USF2 was linked to rheumatoid arthritis that is refractory to treatment through upregulation of proinflammatory cytokines [30], while stabilization of USF2 protein by CDK5-dependent phosphorylation promotes prostate cancer proliferation [36]. Additionally, differential binding of USF1 and USF2 to the plasminogen activator inhibitor type-1 promoter regulates expression during the G_0_ to G_1_ cell cycle transition [67,68]. The C-terminal DNA binding domain of USF1 and USF2 share ~70% sequence identity. However, the N-terminal regions have limited sequence homology [25,27]. Consequently, since these factors are capable of binding the same E-box element [23], either alone or as heterodimers, the specific independent function(s) of these factors has not been fully characterized. Importantly, analysis of the differential effects of USF1 and USF2 for expression of specific genes is complicated by the fact that USF2 is required for USF1 protein expression, at least in the T cell and HEK293T lines. Considering this, it is remarkable that the *USF1* and *USF2* gene knockouts produced notably different transcriptome phenotypes. It will be interesting to characterize the mechanistic differences producing these distinct effects for gene regulation, which may involve differential functions conferred by binding USF2 homodimers relative to USF1/USF2 heterodimers.

We found that USF1 protein levels were significantly reduced upon *USF2* knockout or shRNA knockdown in Jurkat T cells and HEK293T cells. USF2 depletion did not reduce *USF1* mRNA expression in either unstimulated or stimulated T cells (Figure 4), and consequently, the loss of USF2 must cause USF1 protein instability or inhibit translation of its mRNA. Reintroduction of Flag-tagged USF2 in *USF2* null cells restored USF1 protein levels (Figure 4A). We propose that since USF1 and USF2 predominately exist as heterodimers [25], this interaction stabilizes the USF1 protein, and consequently, loss of USF2 may result in USF1 protein instability. We tried to examine this possibility, but this analysis was complicated by the finding that USF1 stability was not enhanced by the proteasome inhibitor MG132, or expression of the C-terminal region of USF2 containing the b-HLH leucine zipper motif (not shown). Nevertheless, such a mechanism regulating stability of heterodimer partners has been described previously for the yeast MATa and α2 proteins, which are mutually stabilized by their heterodimer formation [69].

Our results have revealed novel properties of the ubiquitously expressed USF proteins and clarify their role for regulation of HIV-1 provirus expression. Aberrant expression of *USF2* was previously linked to additional human diseases, including rheumatoid arthritis [30] and cancer [32]. Consequently, a more detailed understanding of the function and regulation of these factors will provide significant insight towards mechanisms that sculpt the transcriptome in response to extracellular signaling. USF proteins have been implicated in the regulation of chromatin modifications and insulator functions [70,71]. Therefore, it will be particularly important to understand the role of USF2 for epigenetic regulation and chromatin organization at the HIV-1 LTR and cellular target genes, as well as the relative contribution of associated factors, including TFII-I for the signal-responsive gene expression.

## Figures and Tables

**Figure 1 viruses-15-01470-f001:**
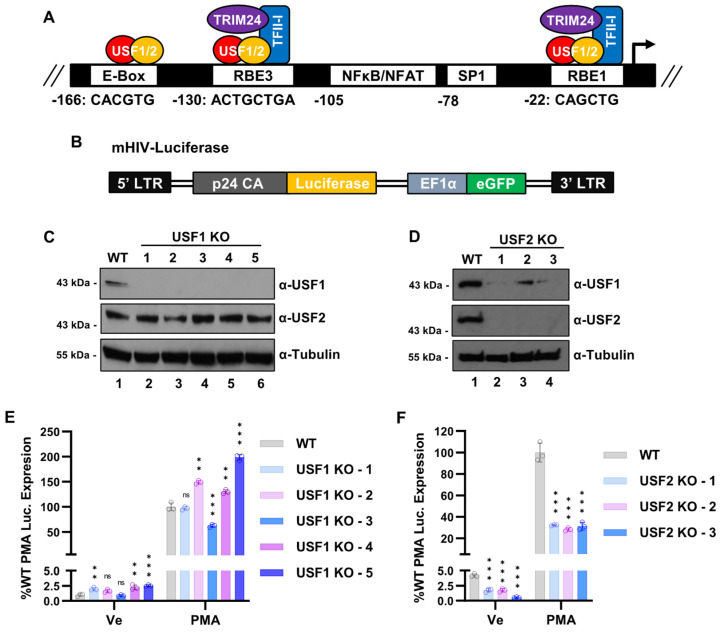
*USF2* knockout inhibits HIV-1 expression in T cells. (**A**): Depiction of the core promoter and proximal enhancer region of the LAI HIV-1 5′ LTR. RBE1 and RBE3 are bound by RBF-2, a complex composed of USF1/2 and TFII-I. TFII-I directly interacts with and recruits the TRIM24 co-factor to the RBE1/3 elements. (**B**): Representation of the reporter provirus in the Jurkat mHIV-Luciferase cell line. The 5′ LTR controls luciferase expression while an internal EF1α constitutive promoter expresses GFP. (**C**,**D**): *USF1* (**C**) and *USF2* (**D**) KO cell lines were produced using CRISPR-Cas9 in the Jurkat mHIV-Luciferase parental cell line. Immunoblotting of cellular lysates was performed using antibodies against USF1, USF2, and Tubulin. (**E**): Wildtype or *USF1* KO mHIV-Luciferase cells were left untreated (Ve, DMSO) or incubated with 5 nM PMA for 4 h prior to measuring luciferase activity (*n* = 3, mean ± SD). (**F**): Same as in (**E**), but the *USF2* KO cell lines were examined (*n* = 3, mean ± SD).

**Figure 2 viruses-15-01470-f002:**
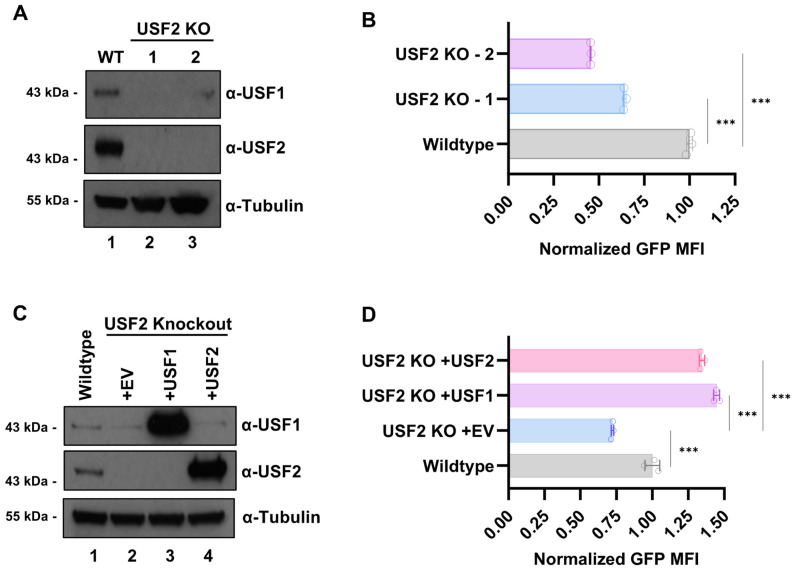
USF1 and USF2 activate HIV-1 LTR transcription. (**A**): *USF2* KO clonal cell lines were generated in HEK293T cells using CRISPR-Cas9 gene editing. The KO was confirmed by immunoblotting whole cell extracts using antibodies against USF1, USF2, and Tubulin. (**B**): Wildtype and *USF2* KO HEK293T cells were transiently co-transfected using an LAI-derived LTR-Tat-IRES-GFP reporter plasmid and constitutive EF1α-RFP vector (Appendix A). The GFP mean fluorescent intensity (MFI) of the RFP expressing population was determined one day post-transfection using flow cytometry and was normalized to the wildtype cells (*n* = 3, mean ± SD). (**C**): HEK293T *USF2* KO cells were transiently transfected with an empty vector (EV) or vectors expressing USF1 or USF2. Whole cell lysates were examined using immunoblotting with antibodies against USF1, USF2, and Tubulin. (**D**): Wildtype and *USF2* KO HEK293T cell lines were co-transfected with an LAI LTR-Tat-IRES-GFP reporter plasmid and either a CMV-EV-EF1α-RFP (EV), CMV-USF1-EF1α-RFP (USF1), or CMV-USF2-EF1α-RFP (USF2) vector (Appendix A). One day post-transfection, the GFP mean fluorescent intensity (MFI) of the RFP expressing population was determined using flow cytometry and normalized to the CMV-EV-EF1α-RFP (EV) transfected wildtype cells (*n* = 3, mean ± SD).

**Figure 3 viruses-15-01470-f003:**
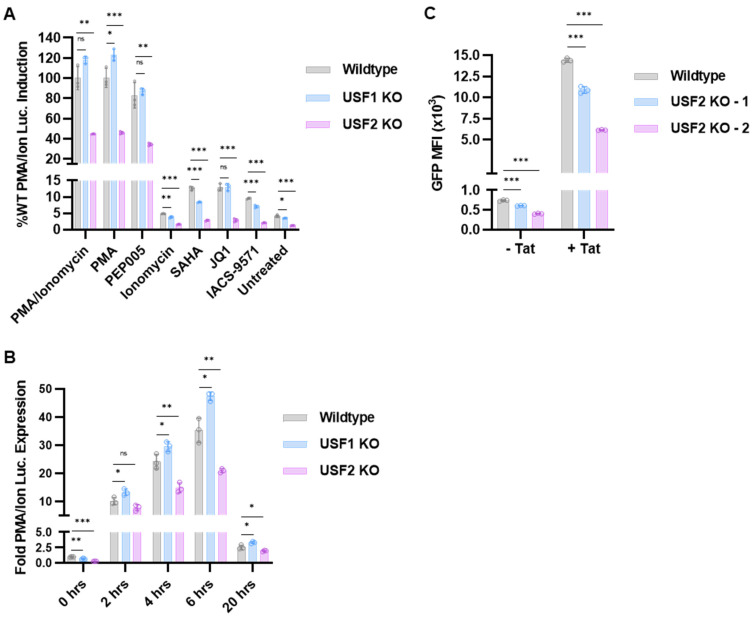
Role of USF for the induction of HIV-1 in response to latency reversing agents and Tat. (**A**): Wildtype, *USF1* KO, or *USF2* KO mHIV-Luciferase cells were left untreated or treated with 5 nM PMA/1 μM ionomycin, 5 nM PMA, 10 nM PEP005, 1 μM ionomycin, 10 μM SAHA, 10 μM JQ1, or 10 μM IACS-9571 for 4 h prior to the luciferase assays (*n* = 3, mean ± SD). (**B**): Wildtype, *USF1* KO, or *USF2* KO mHIV-Luciferase were treated with 5 nM PMA and 1 μM ionomycin for the indicated time (hours), when luciferase expression was measured (*n* = 3, mean ± SD). (**C**): Wildtype or *USF2* KO HEK293T cell lines were co-transfected with the CMV-EV-EF1α-RFP vector and either an LTR-GFP (−Tat) or LTR-Tat-IRES-GFP (+Tat) reporter construct (Appendix A). At one day post-transfection, the mean fluorescent intensity (MFI) of the GFP for the RFP transfected population was determined using flow cytometry (*n* = 3, mean ± SD).

**Figure 4 viruses-15-01470-f004:**
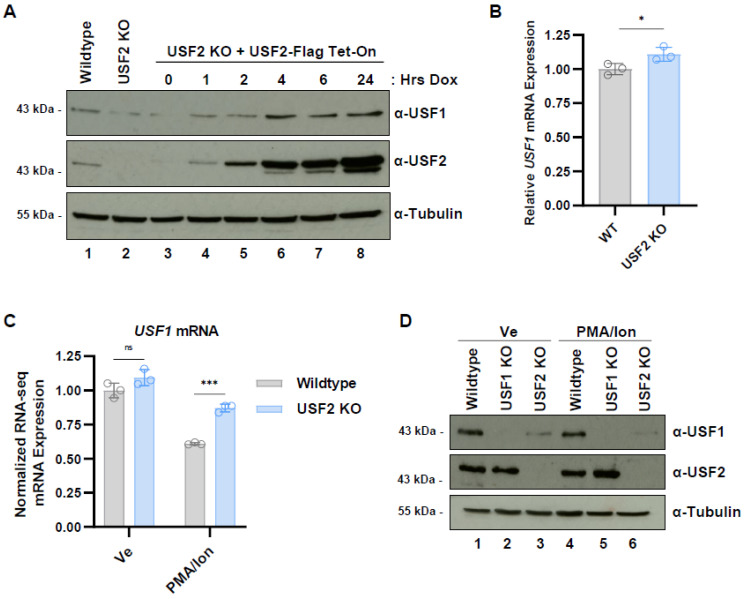
USF1 is stabilized in the presence of USF2. (**A**): *USF2* KO Jurkat T cells were transduced with a doxycycline (Dox)-inducible Flag-tagged USF2 expression vector. Following incubation with 1 μg/mL Dox for the indicated time, whole cell lysates were collected and subjected to immunoblotting using antibodies targeting USF1, USF2, and Tubulin. (**B**): RNA was extracted from WT and *USF2* KO Jurkat cells and subjected to RT-PCR using primers targeting *USF1* mRNA. The *USF1* transcript expression was normalized to *GAPDH* (*n* = 3, mean ± SD). (**C**): Normalized RNA-seq *USF1* mRNA counts of wildtype and *USF2* KO Jurkat cells under basal (Ve) and T cell activated conditions (PMA/Ion) (*n* = 3, mean ± SD). (**D**): Wildtype, *USF1* KO, or *USF2* KO cells were left untreated (Ve, DMSO) or stimulated by treatment with 5 nM PMA and 1 μM ionomycin. Following 4 h, cell lysates were prepared and immunoblotted using antibodies against USF1, USF2, and Tubulin.

**Figure 5 viruses-15-01470-f005:**
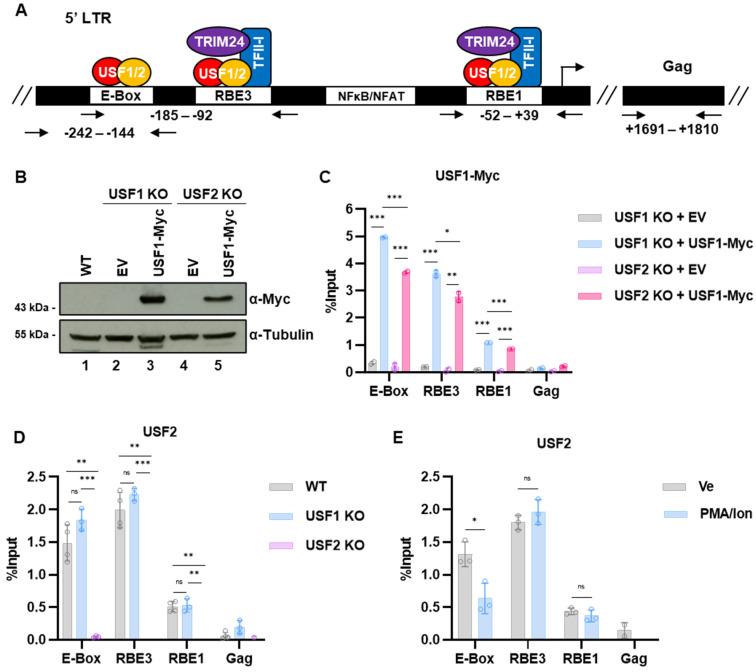
USF1 and USF2 are capable of localizing independently to the LTR. (**A**): Schematic representation of the primer sites used for ChIP-qPCR analysis. (**B**): *USF1* or *USF2* knockout cells were transduced using an empty vector (EV) or Myc-tagged USF1 expression vector. Whole cell lysates were collected, and immunoblotted with antibodies against Myc and Tubulin. (**C**): *USF1* or *USF2* deleted Jurkat T cells transduced with an empty vector (EV) or USF1-Myc expression vector were subjected to ChIP-qPCR using antibodies against Myc. The ChIP-qPCR results were normalized by subtracting values produced with sample paired non-specific IgG (*n* = 2, mean ± SD). (**D**): Wildtype or cells with *USF1* or *USF2* genetic deletions were analyzed by ChIP-qPCR using antibodies targeting endogenous USF2. The ChIP-qPCR results were normalized by subtracting values produced with sample-paired non-specific IgG (*n* = 2–3, mean ± SD). (**E**): Jurkat mHIV-Luc cells were incubated with vehicle control (Ve, DMSO) or 5 nM PMA/1 μM ionomycin for 4 h, after which ChIP-qPCR was performed using antibodies against USF2. The depicted data are qPCR results normalized by subtracting values produced by sample-paired non-specific IgG (*n* = 2–3, mean ± SD).

**Figure 6 viruses-15-01470-f006:**
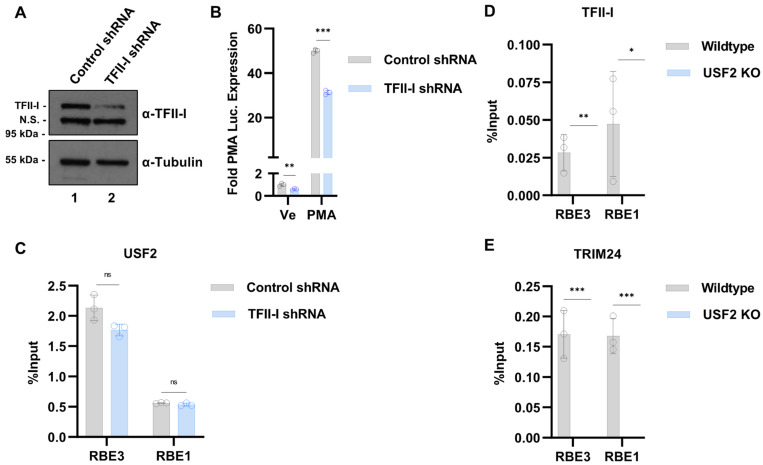
Requirement of USF2 for binding of TFII-I and TRIM24 to the LTR. (**A**): Jurkat mHIV-Luciferase cells were transduced using an LKO control shRNA or a TFII-I targeting shRNA vector. The transduced cells were selected with 7.5 μg/mL puromycin for 8 days after which cell lysates were analyzed by immunoblotting using antibodies against TFII-I and Tubulin. (**B**): shRNA transduced mHIV-Luciferase cells were left unstimulated (Ve, DMSO) or stimulated with 5 nM PMA for 4 h prior to measuring luciferase expression (*n* = 3, mean ± SD). (**C**): Jurkat mHIV-Luciferase cells were transduced with the indicated shRNA, selected with puromycin for 8 days, and subjected to ChIP-qPCR using antibodies against USF2. The data shown are results of qPCR, normalized by subtracting values produced by sample-paired non-specific IgG (*n* = 3, mean ± SD). (**D**): WT or *USF2* KO mHIV-Luciferase cells were subjected to ChIP-qPCR with antibodies targeting TFII-I. The qPCR results were normalized by subtracting values produced by sample-paired non-specific IgG immunoprecipitation (*n* = 2–3, mean ± SD). (**E**): Similar to (**D**), ChIP was performed using antibodies against TRIM24 (*n* = 3, mean ± SD).

**Figure 7 viruses-15-01470-f007:**
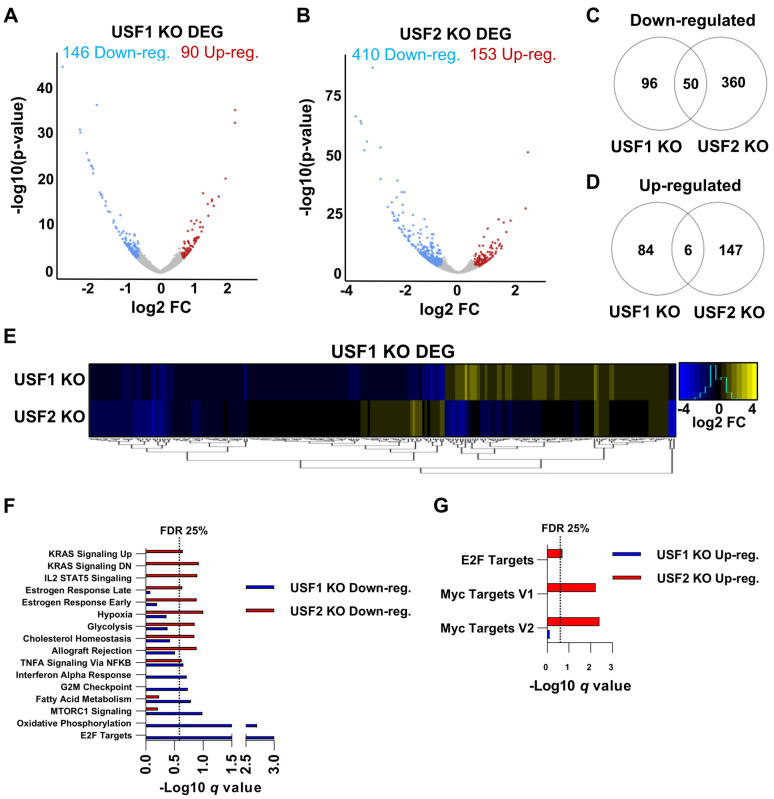
Transcriptomic effects of *USF1* or *USF2* knockouts in unstimulated cells. (**A**,**B**): Volcano plots depicting DESeq2 analysis comparing the wildtype and *USF1* KO (**A**) or *USF2* KO (**B**) Jurkat T cells. The analysis was performed on *n* = 3 RNA-seq samples with significant genes possessing a *p*-value < 0.05 and a fold change in expression > 1.5. (**C**,**D**): Venn diagram depiction of genes significantly down (**C**) or upregulated (**D**) upon *USF1* or *USF2* KO. (**E**): Heatmap depiction of the change in mRNA expression upon *USF1* or *USF2* KO of genes found to be differentially expressed upon *USF1* KO in unstimulated cells. (**F**,**G**): Gene set enrichment (GSEA) of Hallmark gene sets that were downregulated (**F**) or upregulated (**G**) for the *USF1* or *USF2* KO transcriptomes.

**Figure 8 viruses-15-01470-f008:**
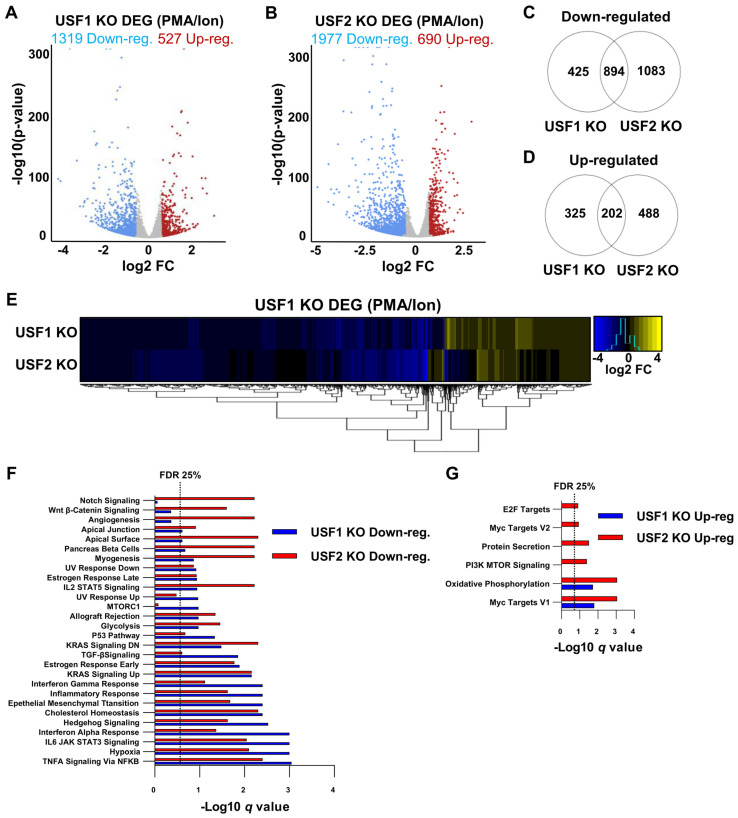
Effect of *USF1* and *USF2* knockout on global gene expression in activated T cells. (**A**,**B**): Volcano plots depicting DESeq2 analysis comparing the WT and *USF1* KO (**A**) or *USF2* KO activated T cells (**B**). Jurkat T cells were stimulated by PMA/ionomycin co-treatment for 4 h prior to RNA extraction. The analysis was performed on *n* = 3 RNA-seq samples with significant genes possessing a *p*-value < 0.05 and a fold change in expression > 1.5. (**C**,**D**): Venn diagrams displaying genes significantly down (**C**) or upregulated (**D**) upon knockout of *USF1* or *USF2* under activating conditions. (**E**) Heatmap depiction of the change in gene expression upon *USF1* KO or *USF2* KO of genes that were differentially expressed upon *USF1* KO under activated T cell conditions. (**F**,**G**): Gene set enrichment (GSEA) of Hallmark gene sets that were downregulated (**F**) or upregulated (**G**) for the USF1/USF2 KO transcriptomes following PMA/ionomycin-mediated T cell activation.

**Figure 9 viruses-15-01470-f009:**
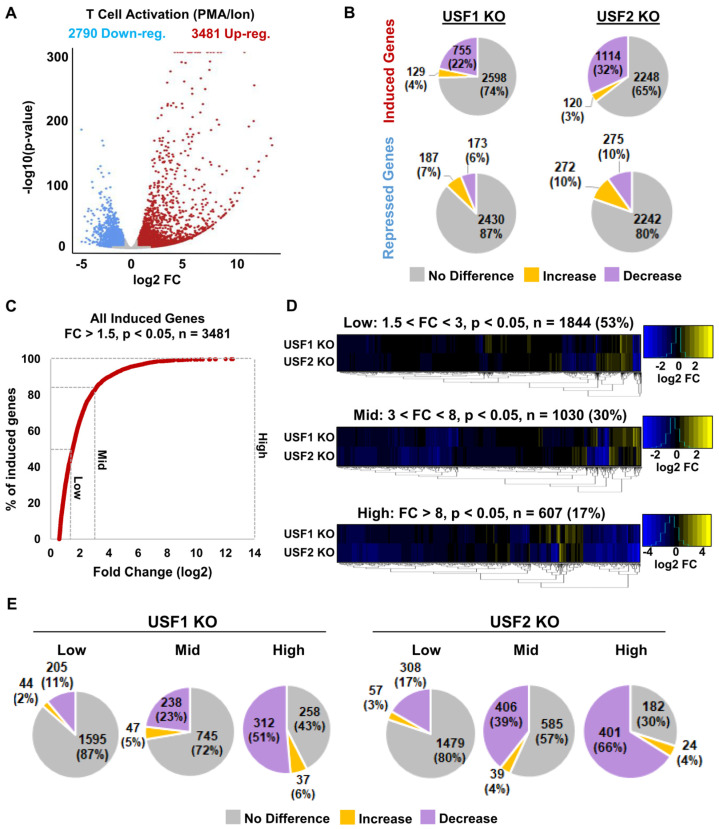
USF1 and USF2 facilitate T cell activation. (**A**): Volcano plot depiction of DESeq2 analysis comparing the transcriptomes of unstimulated and PMA/ionomycin-activated Jurkat T cells. The analysis was performed on *n* = 3 RNA-seq samples with significant genes having a *p*-value < 0.05 and a fold change >1.5. (**B**): Pie charts depicting the effects of *USF1* (left) or *USF2* (right) knockout on T cell activation-induced (top) or -repressed (bottom) genes. (**C**): The number of T cell activation genes that were induced are shown as a percent of the total, given the fold change in expression. (**D**): Heatmap depicting the effects of *USF1* and *USF2* KO on T cell activation-induced genes. The T cell activation-induced genes were stratified based on expression levels into low, mid, and high. (**E**): Pie charts depicting the effects of *USF1* or *USF2* KO on T cell activation genes that were induced at low, mid, and high expression levels.

**Figure 10 viruses-15-01470-f010:**
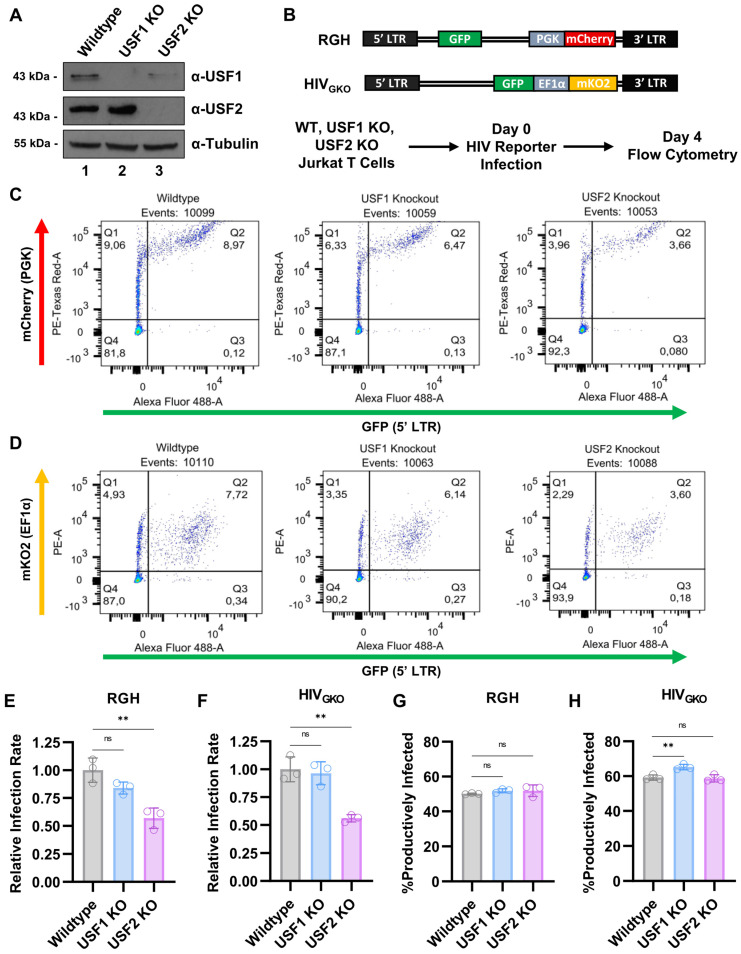
T cells deficient of USF2 are refractory to HIV-1 infection. (**A**): CRISPR-Cas9 was used to produce *USF1* or *USF2* knockouts in Jurkat T cells. Whole cell lysates were subjected to immunoblotting with antibodies against USF1, USF2, and Tubulin. (**B**): Schematic representation of the HIV-1 reporter viruses used to monitor infection. The WT, *USF1* KO, or *USF2* KO Jurkat cells were infected with Red-Green-HIV-1 (RGH) or HIV_GKO_ dual-reporter virus. Four days post-infection, proviral expression was determined using flow cytometry. (**C**,**D**): Representative scatter plots of the RGH (**C**) and HIV_GKO_ (**D**) infected cells, four days post-infection. The uninfected cells that were negative for fluorescence are in Q4, latent infections are located in Q1, the productive infections are in Q2, and noise produced from viral recombination events are shown in Q3. (**E**,**F**): The relative infectivity was determined using flow cytometry four days post-infection of the WT, *USF1* KO, or *USF2* KO Jurkat cells with equivalent amounts of RGH (**E**) or HIV_GKO_ (**F**) virions (*n* = 3, mean ± SD). (**G**,**H**): Similar to (**E**,**F**), the ratio of the productive infections was analyzed using flow cytometry (*n* = 3, mean ± SD).

**Figure 11 viruses-15-01470-f011:**
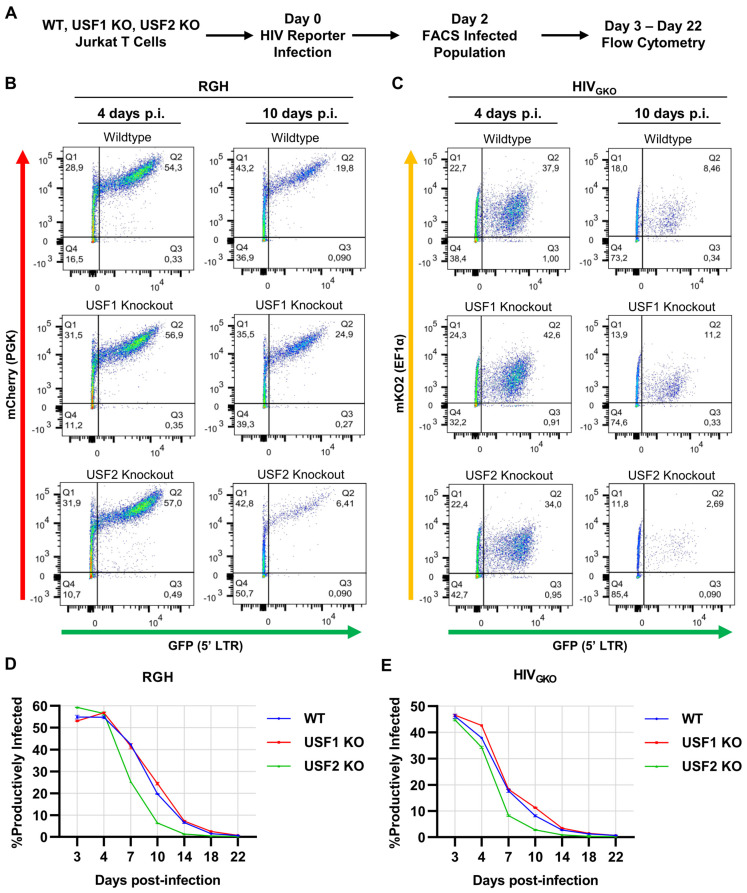
USF2 inhibits the establishment of latency upon HIV-1 infection. (**A**): Schematic representation of the HIV-1 reporter assay. The WT, *USF1* KO, or *USF2* KO Jurkat cells were infected with an RGH or HIV_GKO_ dual-reporter virus at an M.O.I. of ~0.1. Two days post-infection, the infected cells (latent and productive) were isolated using fluorescent activated cell sorting (FACS). HIV-1 expression of the infected population was examined over 22 days using flow cytometry. (**B**,**C**): Representative flow cytometry scatter plot of RGH (**B**) or HIV_GKO_ (**C**) infected Jurkat cells at 4 and 10 days post-infection. (**D**,**E**): Summary of the percent of productive RGH (**D**) or HIV_GKO_ (**E**) infections as determined using flow cytometry at each of the indicated time points (*n* = 2, mean ± SD).

## Data Availability

All the data supporting the findings of this study are available within the article or from the corresponding author upon reasonable request (I. Sadowski, ijs.ubc@gmail.com). All the high-throughput RNA-seq data have been deposited with NCBI GEO, accession GSE227850.

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
