# Peer review of "Upstream Stimulatory Factors Regulate HIV-1 Latency and Are Required for Robust T Cell Activation"

_viruses, 2023, doi:10.3390/v15071470_

Round 1
Reviewer 1 Report
The manuscript by Horvath and Sadowski examines the role of USF-1 and USF-2 in regulating HIV-1 transcription and latency. In general, Jurkat cells lines a
The manuscript by Horvath and Sadowski examines the role of USF-1 and USF-2 in regulating HIV-1 transcription and latency. In general, A CRISPR strategy was used to generate Jurkat cells lines with USF-1 and USF-2 knockout to evaluate whether these factors influence HIV transcription and activation. The data suggest that there is a reduction in transcription and a modest reduction in the ability to induce HIV in lines harboring reporter proviral sequences with latency reversal agents, which may reflect the decrease in baseline transcription. Based on the knockout data and overexpression experiments, the data suggest USF-2, possibly by regulating USF-1 protein levels is influencing HIV transcription. There are also data suggesting knockdown of USF-1 and USF-2 in Jurkat cells alters the transcriptome especially following T activation. The potential impact of these findings is unclear. How these studies extend our understanding of USF-1 and USF-2 on HIV replication and transcription are limited since mechanism is not explored. In addition, critical controls describing phenotypes, growth and survival of the cell lines are lacking and are necessary since these are ubiquitous factor that influence cell proliferation and maturation. Finally, the relevance of studies examining gene expression are unclear since they provide limited insight into intrinsic pathways that may influence HIV replication. Some specific comments are below.
1. The introduction and discussion are unfocused and feel like literature reviews of both the role of USF-1 and USF-2 for HIV transcription and its general function. The hypothesis or questions being addressed are poorly defined. Having a specific model would help direct the reader to the relevance of the reported findings.
2. The effects of USF-2 knockdowns are modest. There is approximately a two fold effect on baseline transcription which is reflected in the “decreased” induction either with LRAs or Tat-mediated transcription. This raises the possibility that the results reflect general decreases in transcriptional potential. Also shown absolute values of different cell lines rather than percentage or normalize values will provide an indication of variation between the different cell lines
3. There is limited information about the cell lines and if multiple cell lines for the knockdowns were used. Were any differences in cell cycle, cell survival, phenotypes observed between the different lines.
4. Single cell transcriptomics are explored suggesting differential expression in Jurkat cells with USF-1 and USF-2 knockdowns; however, how this is relevant to HIV transcription and expression are not explored and potential gene expression networks or signaling cascades are not validated.
5. Broad conclusions about mechanisms are made based on LRAs without validating biochemical mechanisms. For example, it is concluded USF-2 is influencing epigenetic mechanisms based on SAHA treatments, but no additional data which support this claim is provide.
6. A figure showing USF binding sites in the context of the LTR, including the transcriptional start site, TAR and other key binding sites would be helpful to orient the reader as to potential combinatorial transcriptional events.
7. Studies should be performed in cell lines other than Jurkat and although difficult results should be validated in primary T cells.
Author Response
Reviewer #1
Comments and Suggestions for Authors
The manuscript by Horvath and Sadowski examines the role of USF-1 and USF-2 in regulating HIV-1 transcription and latency. In general, Jurkat cells lines a
Comments on the Quality of English Language
The manuscript by Horvath and Sadowski examines the role of USF-1 and USF-2 in regulating HIV-1 transcription and latency. In general, A CRISPR strategy was used to generate Jurkat cells lines with USF-1 and USF-2 knockout to evaluate whether these factors influence HIV transcription and activation. The data suggest that there is a reduction in transcription and a modest reduction in the ability to induce HIV in lines harboring reporter proviral sequences with latency reversal agents, which may reflect the decrease in baseline transcription. Based on the knockout data and overexpression experiments, the data suggest USF-2, possibly by regulating USF-1 protein levels is influencing HIV transcription. There are also data suggesting knockdown of USF-1 and USF-2 in Jurkat cells alters the transcriptome especially following T activation. The potential impact of these findings is unclear. How these studies extend our understanding of USF-1 and USF-2 on HIV replication and transcription are limited since mechanism is not explored. In addition, critical controls describing phenotypes, growth and survival of the cell lines are lacking and are necessary since these are ubiquitous factor that influence cell proliferation and maturation. Finally, the relevance of studies examining gene expression are unclear since they provide limited insight into intrinsic pathways that may influence HIV replication. Some specific comments are below.
- The introduction and discussion are unfocused and feel like literature reviews of both the role of USF-1 and USF-2 for HIV transcription and its general function. The hypothesis or questions being addressed are poorly defined. Having a specific model would help direct the reader to the relevance of the reported findings.
We have modified the Introduction significantly in particular to focus less on general function of USF1 and USF2 and more towards specific effects on HIV-1. We have also provided a stronger description at the end of the Introduction regarding the purpose of the study to examine the effect of USF1 and USF2 on HIV-1 expression by genetic depletion, rather than mutation of cis-elements on the LTR. We have also shortened the Discussion, but have also added clarification of the differential effects we observe for USF1 and USF2 knockouts.
- The effects of USF-2 knockdowns are modest. There is approximately a two fold effect on baseline transcription which is reflected in the “decreased” induction either with LRAs or Tat-mediated transcription. This raises the possibility that the results reflect general decreases in transcriptional potential. Also shown absolute values of different cell lines rather than percentage or normalize values will provide an indication of variation between the different cell lines.
We observe that HIV expression in response to PMA or PMA/Ionomycin treatment is reduced by ~60% upon USF2 KO (Fig. 1F, 2A, 2B). Additionally, cells deficient of USF2 show almost no induction in response to SAHA, JQ1, or IACS-9571 latency reversal agents (Fig. 2A). Nevertheless, the referee notes the important fact that HIV-1 provirus is still significantly response in the USF2 knockout lines, particularly following treatment with PMA or PEP005. We have noted this in the discussion where we note that the LTR is bound by several other factors capable of activating transcription in response to PKC - MAP Kinase signaling.
Regarding normalization of results for expression assays. Luciferase readings give relative light units. This can shift greatly between measurements and it is common place to normalize as a fold change or percentage.
Supporting References:
Barbian, H. J., Seaton, M. S., Narasipura, S. D., Wallace, J., Rajan, R., Sha, B. E. & Al-Harthi, L. β-catenin regulates HIV latency and modulates HIV reactivation. PLoS Pathog 18, e1010354 (2022).
Besnard, E., Hakre, S., Kampmann, M., Lim, H. W., Hosmane, N. N., Martin, A., Bassik, M. C., Verschueren, E., Battivelli, E., Chan, J., Svensson, J. P., Gramatica, A., Conrad, R. J., Ott, M., Greene, W. C., Krogan, N. J., Siliciano, R. F., Weissman, J. S. & Verdin, E. The mTOR Complex Controls HIV Latency. Cell Host Microbe 20, 785–797 (2016).
Horvath, R. M., Brumme, Z. L. & Sadowski, I. Inhibition of the TRIM24 bromodomain reactivates latent HIV-1. Sci Rep 13, 556 (2023).
Horvath, R. M., Dahabieh, M., Malcolm, T. & Sadowski, I. TRIM24 controls induction of latent HIV-1 by stimulating transcriptional elongation. Commun Biol 6, 86 (2023).
Ma, X., Yang, T., Luo, Y., Wu, L., Jiang, Y., Song, Z., Pan, T., Liu, B., Liu, G., Liu, J., Yu, F., He, Z., Zhang, W., Yang, J., Liang, L., Guan, Y., Zhang, X., Li, L., Cai, W., Tang, X., Gao, S., Deng, K. & Zhang, H. TRIM28 promotes HIV-1 latency by SUMOylating CDK9 and inhibiting P-TEFb. Elife 8, e42426 (2019).
Mediouni, S., Chinthalapudi, K., Ekka, M. K., Usui, I., Jablonski, J. A., Clementz, M. A., Mousseau, G., Nowak, J., Macherla, V. R., Beverage, J. N., Esquenazi, E., Baran, P., de Vera, I. M. S., Kojetin, D., Loret, E. P., Nettles, K., Maiti, S., Izard, T. & Valente, S. T. Didehydro-Cortistatin A Inhibits HIV-1 by Specifically Binding to the Unstructured Basic Region of Tat. mBio 10, e02662-18 (2019).
- There is limited information about the cell lines and if multiple cell lines for the knockdowns were used. Were any differences in cell cycle, cell survival, phenotypes observed between the different lines.
Multiple knockout clonal lines were shown for USF1 and USF2 as indicated in Figure 1C, 1D and 2D. Assays were performed in each of the cloned knockout lines as indicated. No change in phenotype was observed in the knockout cell lines, other than effects on HIV-1 expression or infectivity. We have provided additional information on cell growth and viability in Figure S4.
- Single cell transcriptomics are explored suggesting differential expression in Jurkat cells with USF-1 and USF-2 knockdowns; however, how this is relevant to HIV transcription and expression are not explored and potential gene expression networks or signaling cascades are not validated.
RNA Seq analysis was performed on unstimulated or stimulated WT, USF1 or USF2 knockout cell line populations; these results are depicted (now) Figures 7, 8 and 9. HIV utilizes a variety of host cell transcription factors that are activated upon T cell receptor engagement, including NFkB, NFAT, AP1, and Ets. Our work shows that the USF factors are key factors for T cell activation that are likewise co-opted by HIV, and overall, the effect of the USF knockouts on T cell activation response is surprising, but consistent with the view that regulation of HIV-1 provirus expression has evolved to coordinate with activated T cells. Furthermore, previous results indicate HIV-1 infection is enhanced in activated T cells, and our analysis of differentially regulated genes indicates that loss of USF2 would be predicted to impair infectity which we describe in Figures 10 and 11.
I agree that it would be interesting to examine the potential relationships between the USFs and the affected signaling cascades, particularly for example for NFkB and TORC1 signaling, for example, but this seems to be outside the scope of the present manuscript.
- Broad conclusions about mechanisms are made based on LRAs without validating biochemical mechanisms. For example, it is concluded USF-2 is influencing epigenetic mechanisms based on SAHA treatments, but no additional data which support this claim is provide.
Statements including such broad conclusions have been removed from the revised manuscript. It would be interesting to examine effects of the USF deletions on epigenetic alterations at the LTR, and cellular genes, but a proper analysis would require an additional 4 figures at least.
- A figure showing USF binding sites in the context of the LTR, including the transcriptional start site, TAR and other key binding sites would be helpful to orient the reader as to potential combinatorial transcriptional events.
We have added this information as a new Figure panel (Figure 1A).
- Studies should be performed in cell lines other than Jurkat and although difficult results should be validated in primary T cells.
We have added additional results showing effects of the USF2 knockout in the HeLa derived TZM-bl cell line (Figure S2). Additionally, we have examined effects in HEK293 cells, which are an immortalized non-transformed human line. We note that we have also examined effects of the knockouts using multiple models of HIV-1 latency and three different reporter viruses. We agree it would be useful to examine effect of USF2 knockdown in primary T cells. Unfortunately, at the moment, our lab is not able to perform these types of experiments because our human ethics certificate renewal is pending, and I don't expect this to be resolved for several more months.

Reviewer 2 Report
In the manuscript, the authors tested the function of USF1 and USF2 in the context of HIV latency. They found that knockout (KO) of USF2 but not USF1 resulted in a reduction in PMA-triggered reactivation of latent HIV. Despite both being part of the RBF-2 transcription factor that binds to HIV LTR and enhances its transcription, the data in this manuscript suggest that USF1 and USF2 can separately bind to the LTR. Interestingly, they also found that KO of USF2 resulted in the reduction of cellular levels of USF1, but not vice versa. Their finding that USF2 stabilizes USF1 may underlie the observation that KO of USF2 has a more significant effect than KO of USF1 in HIV expression, although their RNA-seq data indicate that both USF1 and USF2 are required for T cell activation. Their viral infection and viral latency establishment assay found that USF2-KO cells are less susceptible to HIV than WT or USF1-KO cells, and get into latency faster once infected. This is an extensive paper covering an important topic (host factors that repress the reactivation of latent HIV), and the major conclusions of this paper are supported by the data. Comments and concerns that should be addressed are:
1. Fig. 1G contradicts with Fig. 3A. The experiments in both figures are highly similar: to ectopically express USF2 in USF2-KO cells, and check if the levels of USF1 restores. However, in Fig. 1G, re-expression of USF2 failed to rescue USF1 levels, while in Fig. 3A, re-expressed USF2 strongly restored USF1 levels to levels even greater than in WT cells. Please comment and discuss and change if appropriate.
2. In lines 457 and 475, the authors stated that “USF2 mediates establishment of HIV latency” and “USF2 promotes the establishment of latency upon HIV-1 infection.” These statements are contradictory to the data. They should be changed to “knock out of USF2 enhances the establishment of HIV latency”, or “USF2 prevents the establishment of HIV latency”.
3. In line 445, the authors stated, “T cells deficient of USF2 are resistant to HIV-1 infection”, while the data clearly show that HIV could still infect the USF2-KO cells. The statement should be changed to “T cells deficient of USF2 are less susceptible to HIV-1 infection”.
Author Response
Reviewer #2
Comments and Suggestions for Authors
In the manuscript, the authors tested the function of USF1 and USF2 in the context of HIV latency. They found that knockout (KO) of USF2 but not USF1 resulted in a reduction in PMA-triggered reactivation of latent HIV. Despite both being part of the RBF-2 transcription factor that binds to HIV LTR and enhances its transcription, the data in this manuscript suggest that USF1 and USF2 can separately bind to the LTR. Interestingly, they also found that KO of USF2 resulted in the reduction of cellular levels of USF1, but not vice versa. Their finding that USF2 stabilizes USF1 may underlie the observation that KO of USF2 has a more significant effect than KO of USF1 in HIV expression, although their RNA-seq data indicate that both USF1 and USF2 are required for T cell activation. Their viral infection and viral latency establishment assay found that USF2-KO cells are less susceptible to HIV than WT or USF1-KO cells, and get into latency faster once infected. This is an extensive paper covering an important topic (host factors that repress the reactivation of latent HIV), and the major conclusions of this paper are supported by the data. Comments and concerns that should be addressed are:
- Fig. 1G contradicts with Fig. 3A. The experiments in both figures are highly similar: to ectopically express USF2 in USF2-KO cells, and check if the levels of USF1 restores. However, in Fig. 1G, re-expression of USF2 failed to rescue USF1 levels, while in Fig. 3A, re-expressed USF2 strongly restored USF1 levels to levels even greater than in WT cells. Please comment and discuss and change if appropriate.
Good catch. In Figure 1G (now Figure 2C), we transiently transfect a USF2 expression vector into HEK293T cells, where only a fraction of the cells express USF2, whereas in Figure 3A (now Figure 4A) USF2 is expressed under control of a doxycycline inducible promoter in a pool of selected transduced cells with the expression construct. We have clarified this in the Figure legends.
- In lines 457 and 475, the authors stated that “USF2 mediates establishment of HIV latency” and “USF2 promotes the establishment of latency upon HIV-1 infection.” These statements are contradictory to the data. They should be changed to “knock out of USF2 enhances the establishment of HIV latency”, or “USF2 prevents the establishment of HIV latency”.
We have made the appropriate correction.
- In line 445, the authors stated, “T cells deficient of USF2 are resistant to HIV-1 infection”, while the data clearly show that HIV could still infect the USF2-KO cells. The statement should be changed to “T cells deficient of USF2 are less susceptible to HIV-1 infection”.
We have changed "resistant" to “refractory” to indicate that USF2 KO cells can be infected by HIV-1 but less efficiently.
Reviewer 3 Report
In this interesting study, Horvath and Sadowski have utilized CRISPR-based and shRNA stable knockdowns to investigate the regulation of global T-cell and HIV proviral gene expression by USF1 and USF2, previously reported to become recruited to certain elements on the HIV promoter as a trimeric complex with TFII-I. Knockdown of USF2 was observed to suppress HIV gene reactivation in Jurkat cells while also abrogating USF1 protein expression. By contrast, knockdown of USF1 did not significantly impact HIV or USF2 expression. In chromatin IP experiments done with Jurkat cells expressing a HIV reporter, that is presumably transcriptionally latent, both USF1 and USF2 were found to be present at the HIV LTR in the unstimulated state. Finally, transcriptomic analysis implicated the cooperativity of these factors in mediating the regulation of cellular genes that are responsive to T-cell activation signals.
This study was well-conducted, the experiments presented have the proper technical controls, the manuscript is well-written and the data presented for the most part support the authors’ findings pertaining to the differential roles of USF1 and 2 in regulating HIV gene expression. Of major concern is the lack of clarity on whether these transcription factors can exert both a repressive and stimulatory function on HIV transcription depending on the cellular activation state. The authors clearly found USF1 and 2 to be occupying the HIV promoter in unstimulated Jurkat cells when viral transcription is presumably minimal. However, no chromatin IP data from the stimulated conditions (PMA or PMA + Ionomycin) are presented to assess whether there is a signal-dependent change in the promoter occupancy of these factors. For instance, while the authors show reactivation of their HIV reporter in response to T-cell stimuli that is suppressed by USF2 depletion, it is not clear whether the reactivation is due to pre-existing promoter-bound USF2 or its enhanced recruitment upon T-cell activation. The authors also indicate that these factors are phosphorylated in response to T-cell activation signals but do not explain if these modifications are important for HIV regulation and whether the modifications can occur while these factors are bound to the promoter. Moreover, might their phosphorylation status determine their activity as repressors or trans-activators?
Major issues that the authors ought to consider addressing in their revision:
11) As described above, to clarify the functional relevance of the presence of USF1 and USF2 at the promoter of the minimally transcribed proviral HIV. Also discuss whether the signal-dependent reactivation of HIV coincides with an enhanced recruitment of these factors or their post-translational modification at the HIV promoter.
2) It would be instructive to the reader if the authors could provide a diagram showing the locations of the putative binding elements for USF1 and 2 at the HIV promoter. It would also be helpful to show the design of the HIV Luciferase reporter construct used to stably generate the Jurkat model. Please clarify whether the reactivation and chromatin IP experiments performed in this study were done with cells containing transcriptionally latent virus. Also show the location(s) of the HIV LTR that was the target of the chromatin IP studies, or the location of the PCR primers used.
3) Figure 2A shows that the PMA only stimulus is just as effective as PMA + Ionomycin at inducing HIV after 4 hrs with almost identical results observed with the knockdowns. Figure 2B shows a transient induction of HIV in response to PMA/Iono. Is this transient pattern also seen with the PMA only stimulation?
4) Interestingly, USF2 depletion abrogates USF1 expression but not vice versa; depletion of USF1 also does not impact the presence of USF2 at the viral LTR. Does depletion of USF2 also affect occupancy of the LTR by TFII-I? When the authors propose that USF2 may bind the LTR as a homodimer, does this imply that the homodimer can form a trimeric complex with TFII-I?
Author Response
Reviewer #3
Comments and Suggestions for Authors
In this interesting study, Horvath and Sadowski have utilized CRISPR-based and shRNA stable knockdowns to investigate the regulation of global T-cell and HIV proviral gene expression by USF1 and USF2, previously reported to become recruited to certain elements on the HIV promoter as a trimeric complex with TFII-I. Knockdown of USF2 was observed to suppress HIV gene reactivation in Jurkat cells while also abrogating USF1 protein expression. By contrast, knockdown of USF1 did not significantly impact HIV or USF2 expression. In chromatin IP experiments done with Jurkat cells expressing a HIV reporter, that is presumably transcriptionally latent, both USF1 and USF2 were found to be present at the HIV LTR in the unstimulated state. Finally, transcriptomic analysis implicated the cooperativity of these factors in mediating the regulation of cellular genes that are responsive to T-cell activation signals.
This study was well-conducted, the experiments presented have the proper technical controls, the manuscript is well-written and the data presented for the most part support the authors’ findings pertaining to the differential roles of USF1 and 2 in regulating HIV gene expression. Of major concern is the lack of clarity on whether these transcription factors can exert both a repressive and stimulatory function on HIV transcription depending on the cellular activation state. The authors clearly found USF1 and 2 to be occupying the HIV promoter in unstimulated Jurkat cells when viral transcription is presumably minimal. However, no chromatin IP data from the stimulated conditions (PMA or PMA + Ionomycin) are presented to assess whether there is a signal-dependent change in the promoter occupancy of these factors. For instance, while the authors show reactivation of their HIV reporter in response to T-cell stimuli that is suppressed by USF2 depletion, it is not clear whether the reactivation is due to pre-existing promoter-bound USF2 or its enhanced recruitment upon T-cell activation. The authors also indicate that these factors are phosphorylated in response to T-cell activation signals but do not explain if these modifications are important for HIV regulation and whether the modifications can occur while these factors are bound to the promoter. Moreover, might their phosphorylation status determine their activity as repressors or trans-activators?
Major issues that the authors ought to consider addressing in their revision:
11) As described above, to clarify the functional relevance of the presence of USF1 and USF2 at the promoter of the minimally transcribed proviral HIV. Also discuss whether the signal-dependent reactivation of HIV coincides with an enhanced recruitment of these factors or their post-translational modification at the HIV promoter.
We have performed ChIP-qPCR of USF2 following PMA/ionomycin mediated activation (Fig. 5) and found that USF2 is constitutively bound to the HIV-1 LTR. This is consistent with previous publications regarding the USFs and other RBF-2 components (TFII-I, and TRIM24).
Supporting References
Bernhard, W., Barreto, K., Raithatha, S. & Sadowski, I. An upstream YY1 binding site on the HIV-1 LTR contributes to latent infection. PLoS One 8, e77052 (2013).
Chen, J., Malcolm, T., Estable, M. C., Roeder, R. G. & Sadowski, I. TFII-I regulates induction of chromosomally integrated human immunodeficiency virus type 1 long terminal repeat in cooperation with USF. J Virol 79, 4396–4406 (2005).
Horvath, R. M., Dahabieh, M., Malcolm, T. & Sadowski, I. TRIM24 controls induction of latent HIV-1 by stimulating transcriptional elongation. Commun Biol 6, 86 (2023).
Malcolm, T., Chen, J., Chang, C. & Sadowski, I. Induction of chromosomally integrated HIV-1 LTR requires RBF-2 (USF/TFII-I) and Ras/MAPK signaling. Virus Genes 35, 215–223 (2007).
Malcolm, T., Kam, J., Pour, P. S. & Sadowski, I. Specific interaction of TFII-I with an upstream element on the HIV-1 LTR regulates induction of latent provirus. FEBS Lett 582, 3903–3908 (2008).
2) It would be instructive to the reader if the authors could provide a diagram showing the locations of the putative binding elements for USF1 and 2 at the HIV promoter. It would also be helpful to show the design of the HIV Luciferase reporter construct used to stably generate the Jurkat model. Please clarify whether the reactivation and chromatin IP experiments performed in this study were done with cells containing transcriptionally latent virus. Also show the location(s) of the HIV LTR that was the target of the chromatin IP studies, or the location of the PCR primers used.
We have added a Figure illustrating the USF/ RBF2 binding sites on the LTR as Figure 1A. Additionally, Figure 1B now shows a schematic representation of the reporter virus in the Jurkat mHIV-Luciferase cell line. Finally, we have added Figure 5A which indicates location of the primers used for qPCR analysis of the ChIP samples.
3) Figure 2A shows that the PMA only stimulus is just as effective as PMA + Ionomycin at inducing HIV after 4 hrs with almost identical results observed with the knockdowns. Figure 2B shows a transient induction of HIV in response to PMA/Iono. Is this transient pattern also seen with the PMA only stimulation?
Yes, this is observed for many latency reversal agents. For reference, see Figure 3 of Horvath et al, 2023 – “Inhibition of the TRIM24 bromodomain reactivates latent HIV-1”.
4) Interestingly, USF2 depletion abrogates USF1 expression but not vice versa; depletion of USF1 also does not impact the presence of USF2 at the viral LTR. Does depletion of USF2 also affect occupancy of the LTR by TFII-I? When the authors propose that USF2 may bind the LTR as a homodimer, does this imply that the homodimer can form a trimeric complex with TFII-I?
We have added additional data to address this issue as Figure 6. Our results show that knockdown of TFII-I expression does not have a significant effect on binding of USF2 to the LTR in vivo (although we cannot achieve complete knockdown in this experiment, Figure 6A). However, loss of USF2 causes complete inhibition of TFII-I binding with the LTR (6D), and also TRIM24 (6E), which is recruited to the LTR by TFII-I. We haven't examined effects of USF1 on binding of other factors by ChIP, but considering that USF2 is capable of binding on its own to the LTR, we think it unlikely that USF1 KO would affect binding of USF2, TFII-I or TRIM24.
The interaction between USF1/ USF2 and TFII-I for binding to the LTR needs further biochemical analysis, as suggested. In previous reports we have shown that TFII-I stimulates binding of USF1/2 to the RBE3 element in vitro, but in these assays, we were not able to detect a heterotrimeric complex bound to DNA (ref 17). It seems, at least in vitro, TFII-I stimulates binding of USF to this element but does not form a stable complex on DNA, consequently we called this effect "quasi-catalytic" (ref 17, 20). I would love to elucidate this biochemically and structurally, but so far, the CIHR has not viewed this as an important objective. I hope that results in this manuscript will reinforce the importance of these interaction for regulation of HIV-1 and global gene expression.
Round 2
Reviewer 1 Report
The authors have made extensive edits based on the original concerns, although suggestions addressing more mechanistic experiments which were dismissed as being beyond the scope of the paper would have improved the broader impact of the paper.
There are several typos and grammatical errors. The paper would benefit from some editing assistance.
Author Response
We thank the referee for all of the comments. We have gone through the manuscript several times and have corrected several spelling and grammatical errors.